# Timing and cell specificity of senescence drives postnatal lung development and injury

Hongwei Yao [1,6] ✉, Joselynn Wallace[2], Abigail L. Peterson[1], Alejandro Scaffa[1], Salu Rizal[1], Katy Hegarty [1], Hajime Maeda[1], Jason L. Chang [1], Nathalie Oulhen[1], Jill A. Kreiling [1], Kelsey E. Huntington [3], Monique E. De Paepe[4], Guilherme Barbosa [1] & Phyllis A. Dennery[1,5,6] ✉

Senescence causes age-related diseases and stress-related injury. Paradoxically, it is also essential for organismal development. Whether senescence contributes to lung development or injury in early life remains unclear. Here, we show that lung senescence occurred at birth and decreased throughout the saccular stage in mice. Reducing senescent cells at this stage disrupted lung development. In mice (<12 h old) exposed to hyperoxia during the saccular stage followed by air recovery until adulthood, lung senescence increased particularly in type II cells and secondary crest myofibroblasts. This peaked during the alveolar stage and was mediated by the p53/p21 pathway. Decreasing senescent cells during the alveolar stage attenuated hyperoxia-induced alveolar and vascular simplification. Conclusively, early programmed senescence orchestrates postnatal lung development whereas later hyperoxia-induced senescence causes lung injury through different mechanisms. This defines the ontogeny of lung senescence and provides an optimal therapeutic window for mitigating neonatal hyperoxic lung injury by inhibiting senescence.

Chronic lung disease affects over 40% of all infants born at ≤29 weeks' gestation[1]. Perinatal and neonatal care has progressed significantly allowing premature infants beyond 22 weeks to survive. Unfortunately, ventilatory support and oxygen therapy used to save them can also disrupt the growth of their distal alveoli and pulmonary microvasculature. This leads to continued dependency on supplemental oxygen beyond 36 weeks corrected gestational age, referred to as bronchopulmonary dysplasia (BPD)[2]. This condition affects 10,000 to 15,000 premature infants annually in the United States. Lung pathology of BPD is characterized by alveolar and vascular simplification as well as dysmorphic vascular development[3–6]. Most BPD survivors

eventually are weaned off oxygen; however, they may show evidence of persistent pulmonary injury and lung function decline as adolescents and adults[7,8]. The mechanisms underlying alveolar simplification in BPD are not fully understood. Although current therapies have improved the survival of premature infants, they have minimally reduced the prevalence of BPD and lung injury[1,9].

Senescence is a term used to describe cells that cease to divide/proliferate and show morphological alterations and pronounced secretory activity. Senescent cells accumulate in aging tissues, which plays pivotal roles in organismal aging and in the pathogenesis of aging-related diseases[10]. Accumulating evidence shows that cellular

[1]Department of Molecular Biology, Cell Biology & Biochemistry, Division of Biology and Medicine, Brown University, Providence, RI 02912, USA. [2]Center for Computational Biology of Human Disease and Center for Computation and Visualization, Brown University, Providence, RI 02912, USA. [3]Department of Pathology and Laboratory Medicine, Warren Alpert Medical School, Brown University, Providence, RI 02903, USA. [4]Department of Pathology, Women and Infants Hospital, Providence, RI 02905, USA. [5]Department of Pediatrics, Warren Alpert Medical School of Brown University, Providence, RI 02903, USA. [6]These authors jointly supervised this work: Hongwei Yao, Phyllis A. Dennery. ✉e-mail: hongwei_yao@brown.edu; phyllis_dennery@brown.edu

senescence also contributes to the development of organs, including limbs, hindbrain roofplate, mesonephros, neural tube, endolymphatic sac, pharyngeal arches, and gut endoderm in humans, naked mole rats, and mice[11–13]. It is unknown whether senescence participates in lung development. We and others have reported that hyperoxia causes senescence in cultured lung fibroblasts, epithelial cells, and fetal airway smooth muscle cells[14–17]. Whether hyperoxia-induced senescence contributes to the pathogenesis of BPD remains unknown. We hypothesized that senescence participates in postnatal lung development, and that this is dysregulated and enhanced with hyperoxic exposure leading to alveolar simplification. To test this hypothesis, we exposed newborn mice to hyperoxia in the saccular stage (postnatal day 0–3, pnd 0–3) to mimic lung injury in BPD. Some mice were then allowed to recover in room air until adulthood, because short-term neonatal hyperoxic exposure has long-term effects on lung function and causes persistent alveolar simplification in mice[18,19]. We assessed the ontogeny of cellular senescence at different stages of postnatal lung development to study whether senescence participates in normal lung alveolarization and maturation. In addition, lung senescence was assessed at different developmental time points after neonatal hyperoxia to determine whether hyperoxia modifies this ontogeny. Furthermore, we employed p21 (Cdkn1a) knockout mice and a selective p21 inhibitor UC2288 to determine the pathway underlying senescence during postnatal lung development and after neonatal hyperoxia. Finally, we reduced lung senescent cells using the senolytic cocktail quercetin/dasatinib at different developmental stages to evaluate the role of senescence in postnatal lung development and hyperoxia-induced lung injury. Here, we show that early programmed senescence

orchestrates postnatal lung development. In contrast, neonatal hyperoxia-induced senescence causes lung alveolar and vascular simplification.

## Results

### Neonatal hyperoxia increases senescence in the alveolar stage

Senescence-associated β-galactosidase (SA-β-gal) activity is a commonly used biomarker for senescence and is detectable at pH 6.0 in senescent cells. Thus, we measured SA-β-gal activity at pnd3, pnd7, pnd10 and pnd60 in lungs of mice exposed to hyperoxia for 3 days as neonates. These time points were chosen to represent the end of saccular stage (pnd3), peak of alveolarization (pnd7-pnd10), and adulthood (pnd60). In normoxia, lung SA-β-gal activity was reduced at pnd7, pnd10 and pnd60 as compared to pnd3 (Fig. 1a, b). Neonatal hyperoxia for 3 days significantly increased lung SA-β-gal activity at both pnd7 and pnd10 (Fig. 1a, b). Loss of nuclear lamin b1 is also considered as a biomarker of senescence[20,21]. In air-exposed control mice, the number of lung cells lacking nuclear lamin b1 was reduced at pnd7, pnd10 and pnd60, compared to those at pnd3 (Fig. 1c, d). Neonatal hyperoxia caused loss of nuclear lamin b1 in mouse lungs at pnd7 (Fig. 1c, d). However, neonatal hyperoxia did not affect the expression of nuclear lamin b1 at pnd3, pnd10, or pnd60 in mice, compared to corresponding air groups (Fig. 1c, d). Increased nuclear lamin b1 loss was also observed in the lungs of premature human infants requiring mechanic ventilation compared to control subjects (Supplementary Fig. 1).

Senescence is inversely related to proliferation[22]. Thus, we determined whether neonatal hyperoxia alters proliferation in lung

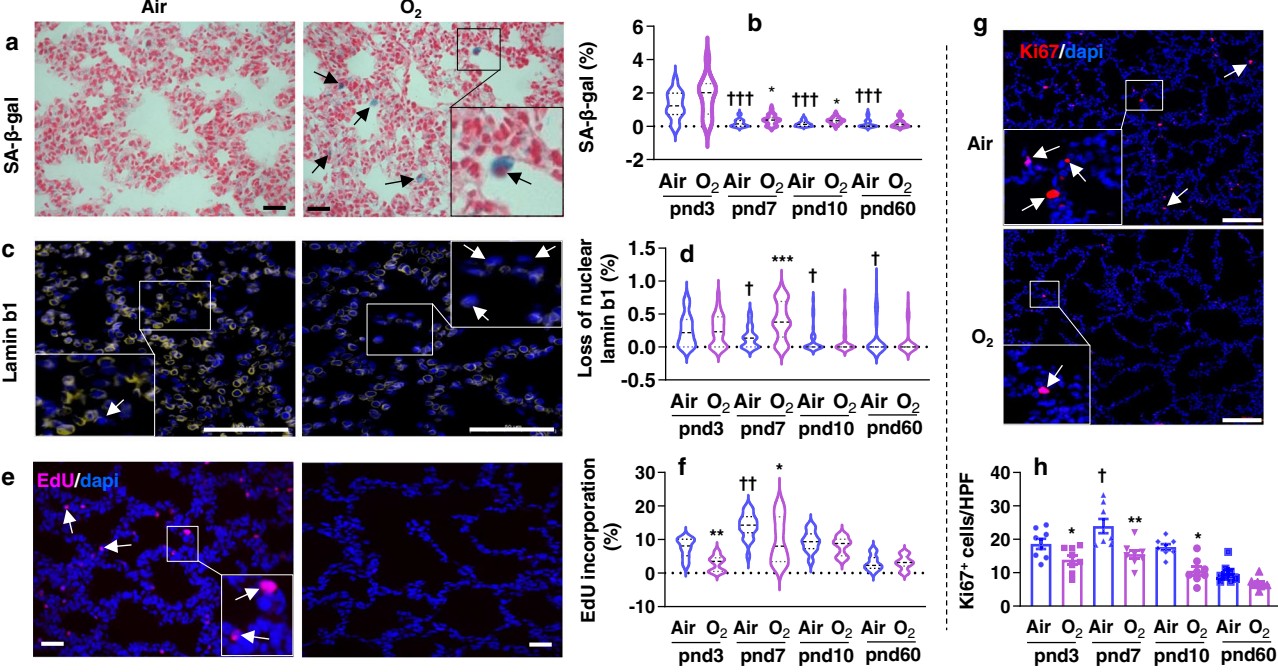

**Fig. 1 | Lung senescence is observed in neonatal hyperoxia-exposed mice.**
C57BL/6 J neonatal mice (<12 h old) were exposed to air or hyperoxia (>95% O₂) for 3 days. Some mice were then allowed to recover in room air until pnd60 (**a–d**, **g**, **h**). **a** SA-β-gal activity was detected by cytochemistry using X-gal as a substrate, and nuclei were stained with nuclear fast red in mouse lungs. Representative images of SA-β-gal staining in mouse lungs at pnd7. Arrows denote SA-β-gal positive cells (blue). **b** Cells positive for SA-β-gal were counted and normalized to total numbers of nuclei in the lung. **c** Immunofluorescence was performed to determine expression of lamin b1. Representative images of lamin b1 staining in mouse lungs at pnd7. Arrows denote cells lacking nuclear lamin b1. **d** Cells lacking nuclear lamin b1 were counted and normalized to total numbers of nuclei in the lung. **e**, **f** Mice were intraperitoneally injected with EdU before sacrifice, and lung tissues were stained

using the Click-iT EdU Proliferation Kit for proliferation. **e** Representative images of EdU staining as shown in red in mouse lungs at pnd3. **f** EdU positive cells were counted and normalized to total numbers of nuclei detected by DAPI in the lung. **g**, **h** Immunofluorescence was performed to detect Ki67 expression in mice exposed to hyperoxia as neonates. **g** Representative images of Ki67 staining as shown in red in mouse lungs at pnd3. **h** Numbers of Ki67 positive cells were counted in 3 randomly selected high-power fields (HPF) for each sample in the lung. Bar size: 25 μm in panels (**a**, **c**, **e**), and 100 μm in panel **g**. Data are expressed as mean ± SEM. N = 8–10 mice per group. Source data are provided as a Source Data file. One-way ANOVA followed by Tukey post-test was used for multiple comparisons. †P < 0.05, †††P < 0.001 vs air group at pnd3; *P < 0.05, **P < 0.01, ***P < 0.001 vs corresponding air group.

cells. Mice were injected with 5-ethynyl-2´-deoxyuridine (EdU) before sacrificing, and incorporated EdU into newly synthesized DNA in proliferative cells was measured using a Click-iT EdU Cell Proliferation kit. We also performed Ki67 staining to detect proliferation in all stages of cell division (except G0). As shown in Fig. 1e–h, cell proliferation was increased at pnd7 as compared to pnd3 under normoxia, and then reduced afterwards. Neonatal hyperoxia significantly reduced proliferation in lung cells at pnd3 and pnd7. The percentage of proliferative cells was not significantly altered by hyperoxic exposure at pnd60 after air recovery (Fig. 1f–h). All these results suggest that lung senescence occurs in newborn mice, is then dramatically reduced during the saccular stage, and that neonatal hyperoxia increases lung cellular senescence at the accelerated phase of alveolarization.

### No sex difference in hyperoxia-induced lung cellular senescence
Male sex is associated with increased severity of BPD in humans and hyperoxic lung injury in mice[23,24]. Therefore, we compared lung senescence between male and female mice exposed to hyperoxia as neonates. As shown in Supplementary Fig. 2, neonatal hyperoxia caused similar levels of lung cellular senescence in both male and female mice, as indicated by increased SA-β-gal activity, nuclear loss of lamin b1, and reduced proliferation. There were no significant differences in SA-β-gal activity, nuclear loss of lamin b1, or reduced proliferation between male and female mice exposed to hyperoxia as neonates (Supplementary Fig. 2). These results demonstrate that there is no sex difference in hyperoxia-induced lung cellular senescence.

### Neonatal hyperoxia increases lung p21 expression and p53 phosphorylation
Although senescence can be mediated by the p53/p21and p16/Rb pathways, p21 is a main mediator of embryonic senescence[12,13,25]. Thus, we first determined lung p21 expression by qRT-PCR and immunofluorescence in mice exposed to hyperoxia as neonates. As shown in Fig. 2a, hyperoxic exposure significantly increased Cdkn1a (p21) mRNA expression in mouse lungs at pnd3, which remained elevated at pnd7 but less so. Similarly, the number of p21 positive cells was significantly

increased in the lung of mice exposed to hyperoxia as neonates at both pnd3 and pnd7 (Fig. 2b, c). Although levels of total p53 protein in the lung were not altered by neonatal hyperoxia at any postnatal day (Fig. 2d, e), hyperoxia increased phosphorylation of p53 at S18, both at pnd3 and pnd7 (Fig. 2f). In addition, p53 phosphorylation at S18 was significantly increased in cells isolated by flow cytometric cell sorting based on $C_{12}FDG$, a substrate of SA-β-gal, staining from hyperoxia-exposed mice at pnd7 (Fig. 2g). Neonatal hyperoxia did not alter Cdkn2a (p16) gene expression in mouse lungs at any postnatal day (Fig. 2h).

To determine whether p21 and phosphorylation of p53 (S18) positive cells are senescent, we performed dual immunofluorescence to determine their colocalization with lamin b1. The numbers of p21/phosphorylation of p53 (S18) positive cells were increased in hyperoxia at pnd7, and these cells lacked nuclear lamin b1 (Supplementary Fig. 3a–d), suggesting senescence in p21 and phosphorylation of p53 (S18) positive cells.

Increased phosphorylation of p53 is associated with apoptosis in neonatal lungs exposed to hyperoxia[26]. Therefore, we determined the colocalization of phosphorylation of p53 (S18) with biomarkers of apoptosis in the lung of mice exposed to hyperoxia as neonates at pnd7. As shown in Supplementary Fig. 3e–h, neonatal hyperoxia increased expression of cleaved caspase-3 and annexin V proteins compared to air group. Interestingly, no colocalization of phosphorylation of p53 (S18) with cleaved caspase-3 or annexin V was observed in the lung of mice exposed to hyperoxia (Supplementary Fig. 3e–h), suggesting that increased phosphorylation of p53 (S18) and apoptosis occurs in different cells. Altogether, these results indicate that hyperoxia activates the p53/p21 pathway, leading to lung cellular senescence in neonatal mice.

### Neonatal hyperoxia causes lung DNA damage
DNA damage is able to trigger cellular senescence by activating the DNA damage response (DDR)[27]. The DDR signaling or repair proteins are assembled rapidly around damage sites, and form DNA damage foci. DNA damage foci are commonly detected by fluorescence

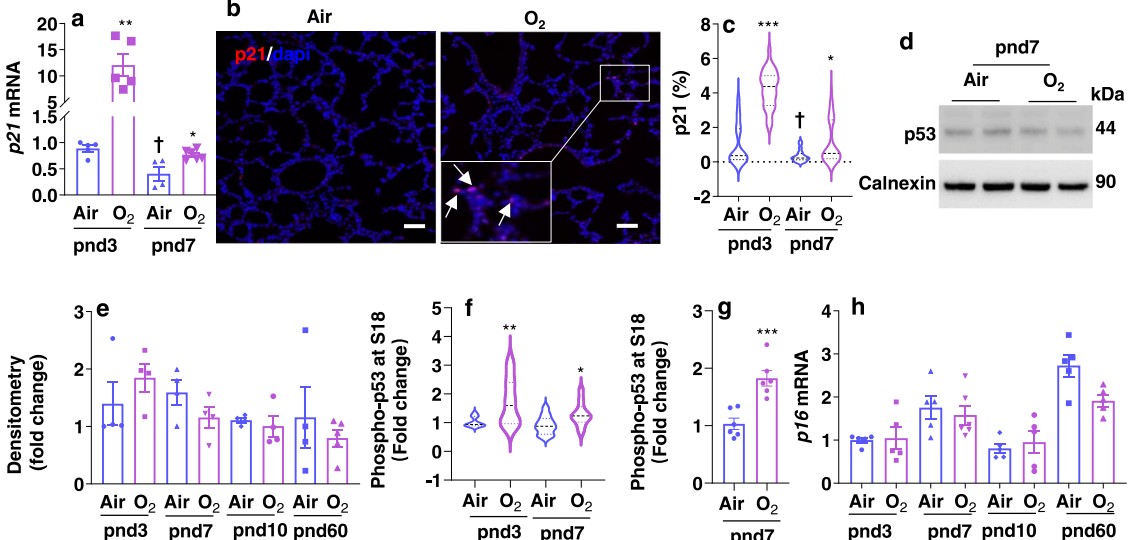

**Fig. 2 | Neonatal hyperoxia increases p21 expression and p53 phosphorylation in mouse lungs.** C57BL/6 J neonatal mice (<12 h old) were exposed to air or hyperoxia (>95% $O_2$) for 3 days. Some mice were then allowed to recover in room air until pnd60. **a** p21 mRNA was measured using qRT-PCR in mouse lungs at pnd3 and pnd7. **b** Immunofluorescence was performed to determine expression of p21 protein. Representative images of p21 staining (red) in mouse lungs at pnd7. Arrows denote p21 positive cells. Bar size: 25 μm. **c** Cells positive for p21 were counted and normalized to total numbers of nuclei in the lung. **d** p53 protein levels were determined by Western blot in mouse lungs at pnd7. **e** Histogram shows p53

protein levels by hyperoxic exposure in mouse lungs at pnd3, pnd7, pnd10 and pnd60. **f** Phosphorylation of p53 at serine 18 was measured by ELISA in mouse lungs at both pnd3 and pnd7. **g** Phosphorylation of p53 at serine 18 was measured by ELISA in $C_{12}FDG$ sorted lung cells at pnd7. **h** p16 mRNA was measured using qRT-PCR in mouse lungs at pnd3, pnd7, pnd10 and pnd60. Data are expressed as mean ± SEM. $N = 4–6$ mice per group. Source data are provided as a Source Data file. †$P < 0.05$ vs air group at pnd3. One-way ANOVA followed by Tukey post-test was used for multiple comparisons (**a, c, e, f, h**), while t-test was used in panel (**g**). *$P < 0.05$, **$P < 0.01$, ***$P < 0.001$ vs corresponding air group.

microscopy detecting the phosphorylated form (Ser139) of the histone variant H2AX (γH2AX) and p53-binding protein 1 (53BP1)[28]. We therefore stained for γH2AX and 53BP1 in the lung of neonatal mice exposed to hyperoxia. As shown in Fig. 3a, b, neonatal hyperoxia increased γH2AX signal at pnd3. However, lung γH2AX signal was not altered at pnd7 or pnd10 in mice exposed to hyperoxia compared to air groups (Fig. 3b). Lung 53BP1 signal was also increased at pnd3 in mice exposed to hyperoxia (Fig. 3c). There were no changes in 53BP1 signal at any other time point (Fig. 3c). We then measured lung 8-oxo-2′-deoxyguanosine (8-oxo-DG), a marker of oxidative DNA damage, in mice exposed to hyperoxia as neonates, since oxidative stress is a known trigger for senescence[29]. As shown in Figs. 3d, 8-oxo-DG signal was increased in lungs of mice exposed to hyperoxia as neonates at both pnd3 and pnd7, suggesting that this oxidative signal may trigger lung cellular senescence.

## Neonatal hyperoxia induces senescence-associated secretory phenotype (SASP)

SASP is associated with senescent cells wherein those cells secrete high levels of inflammatory cytokines, immune modulators, growth factors, and proteases. Therefore, we first revisited our recent published data on single-cell RNA sequencing (scRNA-seq) in mouse lung at pnd7[19]. There were no significant changes in gene expression of proteases (Mmp2, Mmp8, Mmp11, Mmp12, Mmp14, Mmp19, Mmp23, and Mmp25), chemokines (Cxcl1, Cxcl2, Cxcl10, Cxcl14, and Cxcl15), chemokine receptors (Cxcr2 and Cxcr4), or cytokines (Il1α, Il1β, Il6, Il10rα, Il10rβ, and Tnf superfamily) between air and hyperoxia exposed groups (Fig. 3e) in this dataset. We then performed scRNA-seq on lung senescent cells that were isolated from hyperoxia-exposed mice at pnd7 by flow cytometric cell sorting based on $C_{12}$FDG staining.

As expected, the expression of senescence biomarkers *p21*, *Plaur* (uPAR)[30] and *B2M* (β2-microglobulin)[31] was significantly increased in $C_{12}$FDG sorted cells compared to cells from air-exposed lungs (Fig. 3e; Supplementary Data 1). Furthermore, there was a significant increase in expression of *Mmp19*, *Cxcl2*, *Cxcr4*, *Tnfrsf1b*, and *Tnfaip2* genes in $C_{12}$FDG sorted cells at pnd7 (Fig. 3e; Supplementary Data 1). Finally, we performed a commercially available Luminex assay to evaluate levels of selected proteins encoded by the above genes, including cytokines (IL-1α, IL-1β, IL-10, Tnf-α, Tnfrsf1b, and Tnfsf12), chemokines (Cxcl2 and Cxcl12), proteases (MMP-8 and MMP12) and a protease inhibitor (PAI-1). As shown in Fig. 3f, levels of IL-1α, IL-1β, Cxcl2, Cxcl12 and Tnfrsf1b were significantly increased in the lung of mice exposed to hyperoxia compared to air. This suggests that SASP factors are induced in senescent cells from neonatal mice exposed to hyperoxia.

## Hyperoxia causes cell-specific senescence in the lung

The lung is composed of over 40 cell types. To determine whether hyperoxia mediated lung senescence is cell-specific, we analyzed the scRNA-seq data in $C_{12}$FDG sorted cells at pnd7. After filtering and cell state annotations in Seurat, there were 493 $C_{12}$FDG positive senescent cells of various types, including macrophages (454 cells), type II cells (8 cells), type I cells (8 cells), mesenchymal cells (5 cells), endothelial cells (3 cells), and 15 other immune cells. Moreover, there was a significant difference in SASP gene expression among different types of senescent cells in the lung of hyperoxia-exposed mice at pnd7 (Supplementary Fig. 4a), which is confirmed by the Wilcoxon rank sum test comparing gene expression across the Seurat clusters of the senescent cells (Supplementary Data 2–3).

Since the majority of $C_{12}$FDG positive cells are macrophages, we further investigated the characteristics of these macrophages and

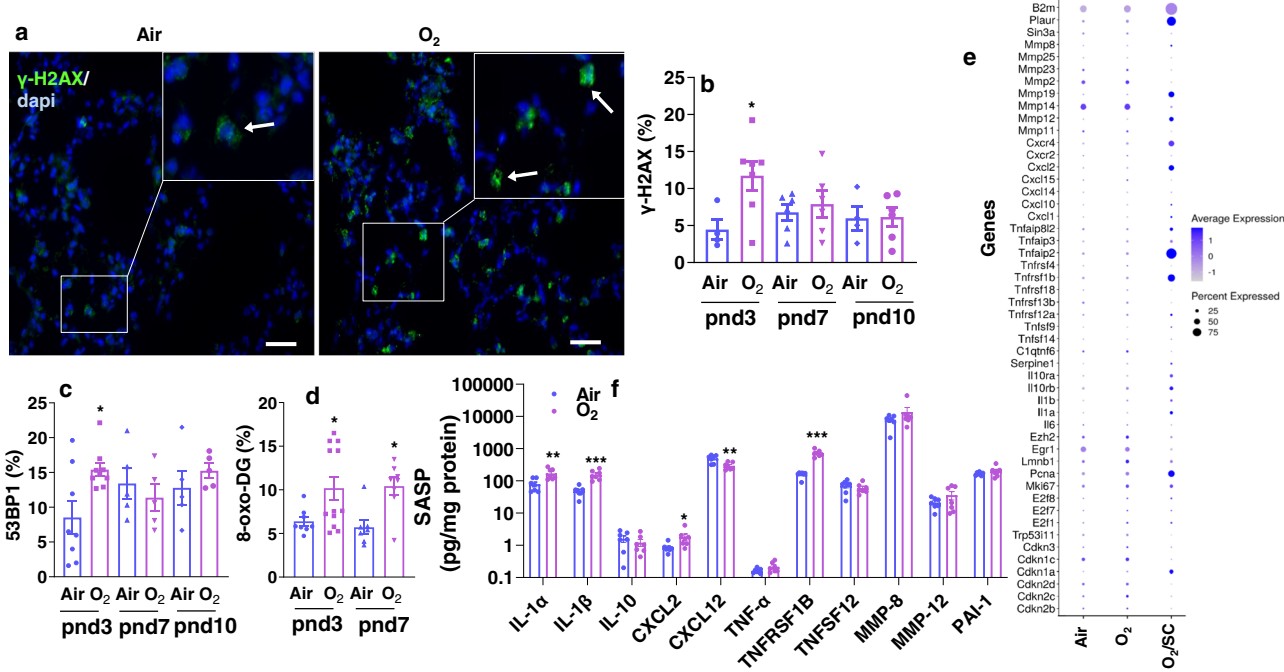

**Fig. 3 | Neonatal hyperoxia increases DNA damage and SASP expression in mouse lungs.** C57BL/6 J neonatal mice (<12 h old) were exposed to air or hyperoxia (95% $O_2$) for 3 days. Some mice were then allowed to recover in room air until pnd10. **a** Immunofluorescence was performed to determine expression of γH2AX. Representative images of γH2AX staining as shown in green in mouse lungs at pnd3. Arrows denote γH2AX foci. Bar size: 25 μm. **b** Cells positive for γH2AX were counted and normalized into total numbers of nuclei detected by DAPI in mouse lungs at pnd3, pnd7, and pnd10. **c** Immunofluorescence was performed to determine expression of 53BP1. Cells positive for 53BP1 were counted and normalized into total numbers of nuclei detected by DAPI in the lungs. **d** Immunofluorescence was performed to determine expression of 8-oxo-DG. Cells positive for 8-oxo-DG were counted and normalized into total numbers of nuclei detected by DAPI in mouse lungs at pnd3 and pnd7. **e** scRNA-seq was performed in lung single cell suspensions without $C_{12}$FDG sorting from air and hyperoxia ($O_2$)-exposed mice[19], as well as in $C_{12}$FDG-sorted lung single cells from hyperoxia group ($O_2$/SC) at pnd7. Dot size is proportional to the percentage of cells expressing each gene. **f** Luminex assay was performed to evaluate the levels of SASP factors in whole lungs at pnd7. Data are expressed as mean ± SEM. N = 5–7 mice per group. Source data are provided as a Source Data file. One-way ANOVA followed by Tukey post-test was used for multiple comparisons. *P < 0.05, **P < 0.01, ***P < 0.001 vs corresponding air group.

identified six Seurat clusters, including clusters 13, 15, 16, 17, 21, and 23 that contained these cells. Macrophages may change their polarization status once they acquire a senescence-like phenotype or when exposed to SASP factors[32]. Thus, we wanted to evaluate polarization phenotype of these clusters using M1 markers (Il1α, Il1β, Il6, Nos2, Tlr2, Tlr4, Cd80 and Cd86) and M2 markers (Csf1r, Mrc1, Pparg, Arg1, Cd163, and Pdcd1lg2). As shown in Supplementary Data 4, clusters 13 and 15 highly expressed M2 markers, while cluster 23 exhibited a mixed M1/M2 phenotype. Other clusters did not specifically express M1 or M2 markers (clusters 16 and 21) or showed reduced expression of a M1 marker Cd86 (cluster 17). These results indicate that specific macrophage subpopulations change their polarization status towards M2 phenotype when expressing senescence markers.

Senescent macrophages display a deficit in phagocytosis, whereas healthy macrophages are able to clear senescent cells by phagocytosis[33–35]. Therefore, we evaluated whether the macrophages recovered using the scRNA-seq after $C_{12}$FDG sorting lost their markers of phagocytosis. Interestingly, as shown in Supplementary Data 3, 5, 6, specific cell clusters (e.g., cluster 13) co-expressed biomarker genes of type II cells and SASP factors as well as genes regulating phagocytosis suggesting that healthy macrophages are engulfing type II cells. Further exploration with immunohistochemistry revealed co-localization of a macrophage marker F-40/80 with pro-SPC in the lung of mice exposed to hyperoxia as neonates at pnd7 (Supplementary Fig. 4b). This suggests that certain macrophages engulf and ingest senescent type II cells in the lung of mice exposed to hyperoxia as neonates.

We next performed immunofluorescence to validate cell-specific senescence in hyperoxia-exposed mouse lungs at pnd7. As shown in Supplementary Fig. 5a, we did not observe p21 colocalization with a type I cell biomarker Hopx in the lung of hyperoxia-exposed mice at pnd7. Similarly, Hopx+ cells did not exclude nuclear lamin b1 in the lung of both air and neonatal hyperoxia-exposed mice at pnd7 (Supplementary Fig. 5b). Furthermore, there was no senescence in endothelial cells, as indicated by absence of colocalization of p21 with an endothelial cell biomarker vWF, as well as absence of lamin b1 exclusion in vWF+ cells, in the lung of mice exposed to hyperoxia as neonates at pnd7 (Supplementary Fig. 5c, d). This confirms that there was no senescence in lung type I or endothelial cells in the lung of neonatal hyperoxia-exposed mice.

We then carried out immunofluorescence using a type II cell biomarker pro-SPC or a mesenchymal cell biomarker vimentin in combination with lamin b1 and p21 at pnd7. As shown in Fig. 4, hyperoxic exposure caused nuclear loss of lamin b1 in pro-SPC+ cells. The number of p21+/pro-SPC+ cells was significantly increased in the lung of mice exposed to hyperoxia. We also observed increased p21 signal and loss of nuclear lamin b1 in vimentin+ mesenchymal cells (Fig. 5a–d). Secondary crest myofibroblasts play important roles in mediating alveologenesis[36]. To study whether these cells are senescent, we performed immunofluorescence of a secondary crest myofibroblast biomarker Pdgfra with lamin b1 and p21. As shown in Fig. 5e–h, neonatal hyperoxia caused nuclear loss of lamin b1 in Pdgfra+ cells. The number of cells colocalizing p21 and Pdgfra was significantly increased in the lung of mice exposed to hyperoxia. Altogether, these results suggest that neonatal hyperoxia causes senescence in lung type II cells and Pdgfra+ secondary crest myofibroblasts.

## Neonatal hyperoxia results in senescence in Sox9 positive cells

The transcription factor Sox9 expressed in the epithelium promotes proper branching morphogenesis and lung repair after injury, and it is drastically reduced in mouse lung after birth[37–40]. We therefore evaluated whether hyperoxia causes senescence in Sox9 positive cells at pnd7. As shown in Supplementary Fig. 6a–f, neonatal hyperoxia increased Sox9 expression but also caused senescence in Sox9+ cells. Interestingly, *Sox9* gene expression exhibited a 58% reduction in senescent type II cells isolated from mice exposed to hyperoxia

compared to those in non-senescent type II cells isolated from air control (Supplementary Fig. 6g). We also detected the colocalization of Sox9 with pro-SPC or Pdgfra in the lung of neonatal hyperoxia-exposed mice at pnd7. As shown in Supplementary Fig. 7a, b, colocalization of Sox9 with pro-SPC was reduced in the lung of mice exposed to hyperoxia as neonates compared to air group. Although hyperoxia reduced Pdgfra expression, there was no colocalization of Sox9 with Pdgfra in the lung of both air and hyperoxia-exposed mice at pnd7 (Supplementary Fig. 7c, d). These results demonstrate that neonatal hyperoxia is linked to senescence in Sox9 positive cells but inhibits Sox9 gene expression in senescent type II cells.

## Senescent type II cells cause lung injury via secretory SASP factors

To determine whether hyperoxia-induced senescence leads to lung injury, we isolated senescent type II cells from mice exposed to hyperoxia as neonates at pnd7, and then cultured them for 24 h. As shown in Fig. 6a, pro-SPC+ cells accounted for approximately 85% of total isolated cells. The SASP factors, including IL-1α, IL-1β, Cxcl2, Cxcl12, and Tnfrsf1b, were significantly increased in the supernatants from these cells compared to air control (Fig. 6b). We then intranasally injected 20 μl of culture media into neonatal mice at pnd7 and pnd10, and measured mean linear intercept (Lm) and radial alveolar count (RAC) in these mice at pnd14. As shown in Fig. 6c, intranasal administration of supernatants from senescent cells significantly increased Lm and reduced RAC. Additionally, vascular simplification was also observed in the mice intranasally administered with supernatants from hyperoxia-induced senescent cells (Fig. 6d). These results demonstrate that senescent cells contribute to hyperoxia-induced alveolar and vascular simplification through secretory SASP factors.

## Treatment with a senolytic cocktail and a selective p21 inhibitor attenuates hyperoxia-induced lung injury

To determine whether reducing senescent cells protects against hyperoxia-induced lung injury, we administered the senolytic cocktail quercetin/dasatinib (2.5 and 5 mg/kg, i.p.) at pnd4 and pnd6 into mice exposed to hyperoxia as neonates. We first determined whether lung senescence markers are reduced by quercetin/dasatinib treatment at pnd7. Increased SA-β-gal seen in neonatal hyperoxia was significantly inhibited by quercetin/dasatinib treatment in a dose-dependent manner (Supplementary Fig. 8a, b). Likewise, administration of quercetin/dasatinib significantly decreased the hyperoxia-induced loss of nuclear lamin b1 in the lung (Supplementary Fig. 8c, d). This suggests that quercetin/dasatinib efficiently reduces lung senescence markers in mice exposed to hyperoxia as neonates.

We next assessed whether treatment with quercetin/dasatinib affects hyperoxia-induced lung injury. At pnd14, lung H&E staining was performed in quercetin/dasatinib-treated mice exposed to hyperoxia as neonates. Neonatal hyperoxia increased the Lm and reduced RAC in mice (Fig. 7a–c). Treatment with quercetin/dasatinib during the saccular stage significantly inhibited the hyperoxia-induced increase in Lm and decrease in RAC in a dose-dependent manner (Fig. 7a–c). Furthermore, neonatal hyperoxia-induced vascular simplification, as indicated by decreased vWF positive blood vessels, was significantly attenuated by quercetin/dasatinib treatment (Fig. 7d, e).

Since hyperoxia increased p21, we determined the role of p21 in hyperoxia-mediated senescence and lung injury. We administered a specific p21 inhibitor UC2288 (2 and 5 mg/kg) into mice at pnd4 and pn6. As shown in Fig. 8a–d, UC2288 administration significantly inhibited *p21* gene expression, SA-β-gal activity and loss of nuclear lamin b1 in the lung in response to hyperoxia exposure. Hyperoxia-induced alveolar simplification was also attenuated by UC2288 in a dose-dependent manner (Fig. 8e, f). Additionally, hyperoxia-induced reduction in vWF positive blood vessels was significantly restored by UC2288 treatment (5 mg/kg) (Fig. 8g). Altogether, reducing

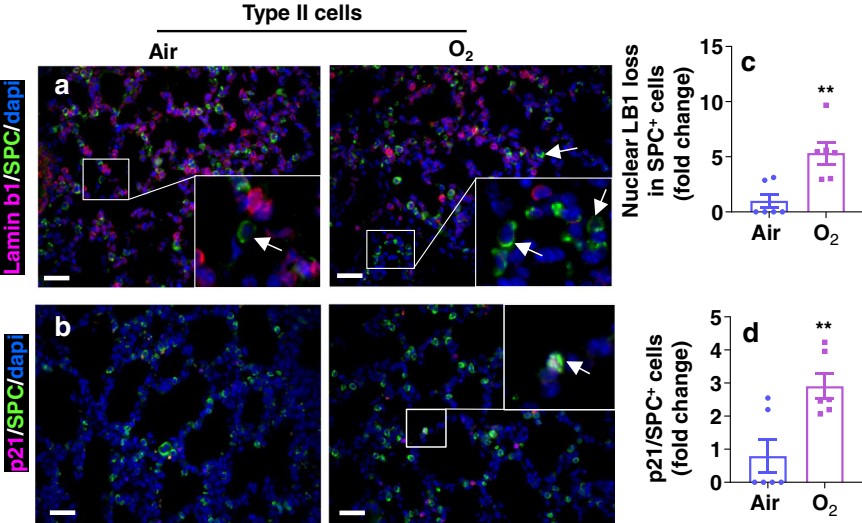

**Fig. 4 | Neonatal hyperoxia increases senescence markers in lung type II cells.** C57BL/6 J neonatal mice (<12 h old) were exposed to air or hyperoxia (95% O₂) for 3 days followed by air recovery until pnd7. **a, b** Immunofluorescence was performed to determine co-localization of p21 and loss of nuclear lamin b1 with pro-SPC (SPC) in the lung. Cells lacking nuclear LB1 but positive for SPC (**c**), or p21⁺/SPC⁺ (**d**) were counted and normalized to total numbers of nuclei. Arrows denote nuclear loss of lamin b1 in SPC⁺ (**a**), or p21⁺/SPC⁺ (**b**) cells. Bar size: 50 μm. Data are expressed as mean ± SEM. N = 5–7 mice per group. Source data are provided as a Source Data file. t-test was used for comparison. **P < 0.01 vs air group.

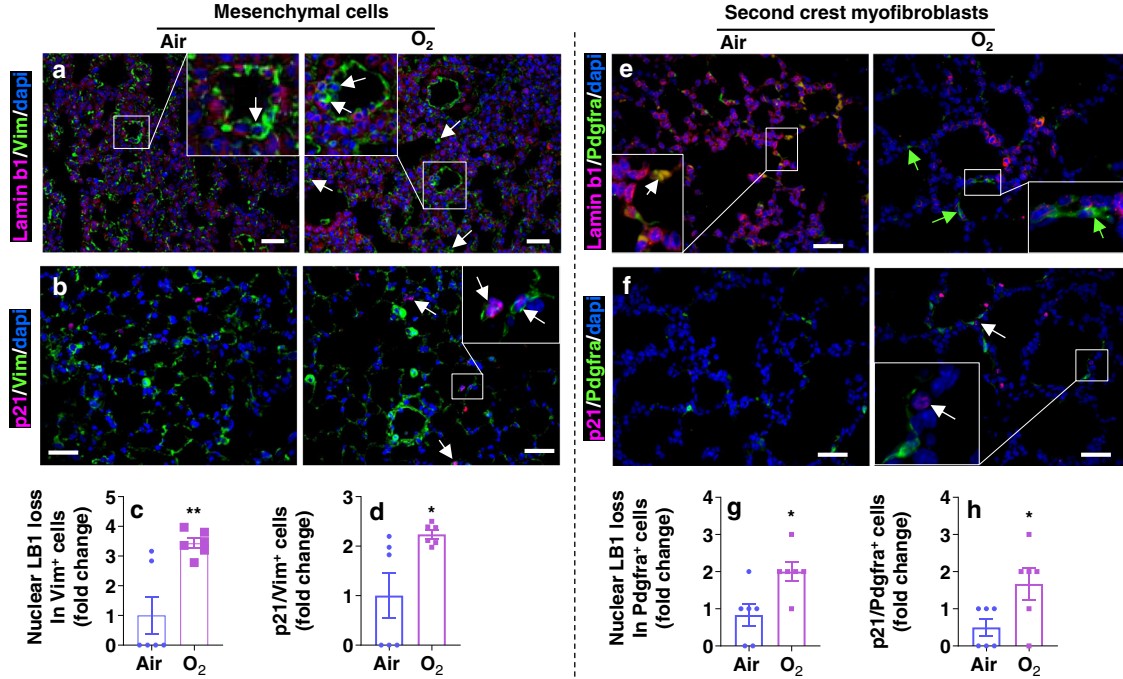

**Fig. 5 | Neonatal hyperoxia increases senescence markers in Pdgfra⁺ mesenchymal cells.** C57BL/6 J neonatal mice (<12 h old) were exposed to air or hyperoxia (95% O₂) for 3 days followed by air recovery until pnd7. **a, b** Immunofluorescence was performed to determine co-localization of p21, and loss of nuclear lamin b1 (LB1) with vimentin in the lung. Cells without nuclear LB1 but positive for vimentin (Vim, **c**), or p21⁺/Vim⁺ (**d**) were counted and normalized to total numbers of nuclei. **e, f** Immunofluorescence was performed to determine co-localization of p21 and loss of nuclear lamin b1 with Pdgfra in the lung. Bar size: 50 μm. Cells lacking nuclear LB1 but positive for Pdgfra (**g**), or p21⁺/Pdgfra⁺ (**h**) were counted and normalized to total numbers of nuclei. Arrows denote nuclear loss of lamin b1 in Vim⁺ (**a**), p21⁺/Vim⁺ (**b**), nuclear loss of lamin b1 in Pdgfra⁺ (**c**), or p21⁺/Pdgfra⁺ (**d**) cells. Data are expressed as mean ± SEM. N = 5–7 mice per group. Source data are provided as a Source Data file. t-test was used for comparison. *P < 0.05, **P < 0.01 vs air group.

senescence markers during the alveolar stage inhibits hyperoxia-induced alveolar and vascular simplification.

**Lung senescent cells are decreased during the saccular stage**
Since lung cellular senescence was reduced in the alveolar stage (pnd7) compared to the saccular stage (pnd3) (Fig. 1), we wanted to evaluate

the dynamic changes in senescence during postnatal saccular stage under normoxic conditions. As shown in Fig. 9a, b, lung SA-β-gal activity peaked at pnd0 and was drastically reduced from pnd0 to pnd3. Similarly, loss of nuclear lamin b1 and *p21* gene expression was also dramatically decreased during the saccular stage of lung development (Fig. 9c, d). There were no changes in *p16* gene expression in

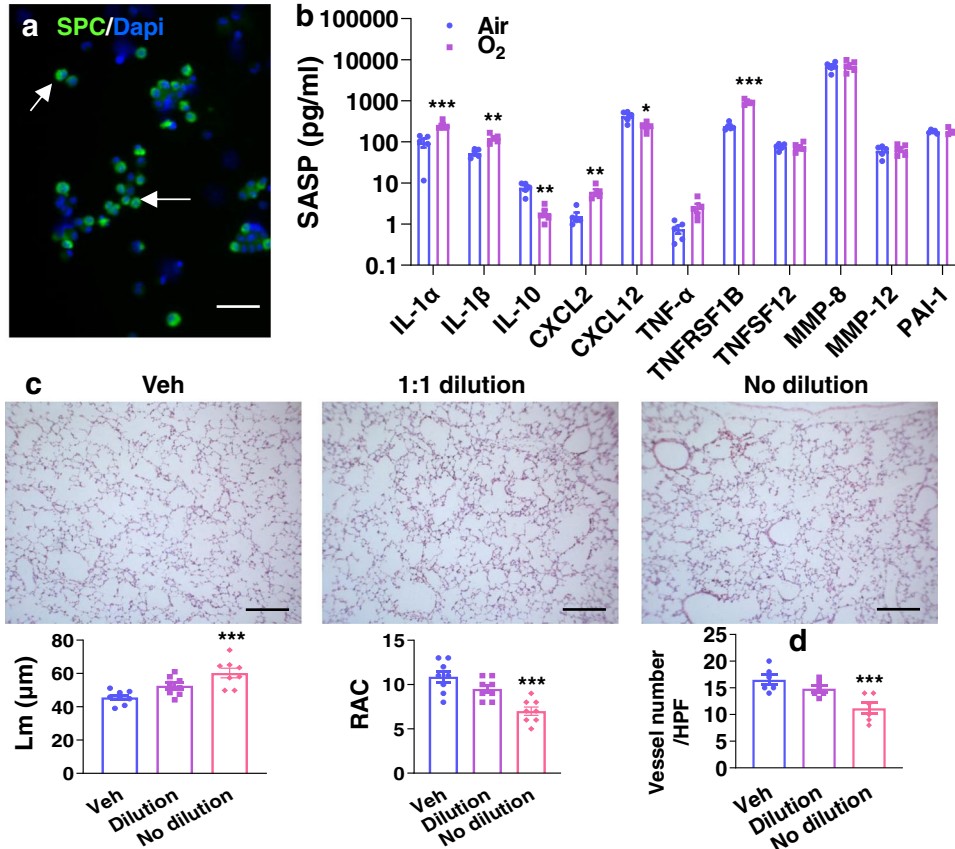

**Fig. 6 | Intranasal administration of supernatants from cultured senescent type II cells causes alveolar and vascular simplification.** C57BL/6 J neonatal mice (<12 h old) were exposed to air or hyperoxia (95% O₂) for 3 days followed by air recovery until pnd7. Senescent type II cells were isolated from these mice through the sorting of pro-SPC and C12FDG positive cells. **a** Immunofluorescence of pro-SPC was performed in these cells after 24 h of culture. Arrows denote pro-SPC positive cells. Bar size: 50 μm. **b** Luminex assay was performed to evaluate the levels of SASP factors in the supernatants. **c, d** C57BL/6 J neonatal mice were intranasally injected with 20 μl of culture medium with or without dilution (1-part medium:1-part saline) of cultured cells at pnd7 and pnd10. H&E and vWF staining was performed to measure mean linear intercept (Lm), radial alveolar count (RAC) and number of blood vessels in the lung at pnd14. Bar size: 100 μm. Data are expressed as mean ± SEM. *N* = 5–6 mice per group. Source data are provided as a Source Data file. One-way ANOVA followed by Tukey post-test was used for multiple comparisons (**c**, **d**), while *t*-test was used in panel (**b**). *P < 0.05, **P < 0.01, ***P < 0.001 vs air (**b**) or vehicle (**c**, **d**).

the lung of the pnd0, pnd1 and pnd3 groups (Fig. 9e). These data indicate that lung senescence is a natural phenomenon in newborn mice, which peaks in the saccular stage of postnatal lung development.

## Characterization of developmentally regulated senescence in the lung

To characterize developmentally regulated senescence in the lung, we first performed immunofluorescence of γH2AX and 53BP1 in the lung during postnatal saccular stage under normoxic condition. As shown in Supplementary Fig. 9a–f, there were no changes in γH2AX or 53BP1 expression in the lung of the pnd0, pnd3 and pnd7 groups. We then evaluated the levels of SASP factors in the lung during postnatal lung development using a Luminex assay. As shown in Supplementary Fig. 9g, there were no changes in IL-1α, IL-1β, Cxcl2, Cxcl12 or Tnfrsf1b in lung homogenates of the pnd0, pnd3 and pnd7 groups under normoxia. The levels of IL-10, TNF-α, MMP-8, and PAI-1 were reduced, while Tnfsf12 and MMP-12 levels were increased at pnd7 compared to pnd0 in the lung (Supplementary Fig. 9g).

Finally, we performed immunofluorescence to detect senescence markers in specific cells. As shown in Supplementary Fig. 10a, b, there were no changes in nuclear lamin b1 loss in pro-SPC positive cells in the pnd0, pnd3 and pnd7 groups under normoxia. The number of cells colocalizing p21 and pro-SPC was also unchanged during the postnatal saccular stage under normoxic condition (Supplementary Fig. 10c, d). Nuclear lamin b1 loss and p21 expression were observed in vimentin

positive cells at pnd0, and this was dramatically decreased from pnd0 to pnd7 (Supplementary Fig. 10e, f). There were no changes in senescence markers in Pdgfra positive cells during postnatal lung development (Supplementary Fig. 10g, h). These findings demonstrate that vimentin⁺ mesenchymal cell senescence during postnatal lung development is not a DNA damage-triggered event.

## Reducing senescent cells during the saccular stage disrupts lung development

To determine whether programmed senescence mediated by p21 orchestrates postnatal lung development, we employed *p21* knockout mice. As shown in Fig. 10a, b, lung cellular senescence was reduced in *p21* knockout mice as compared to WT littermates under normoxic condition at pnd4 and pnd7. Deletion of *p21* reduced numbers of alveoli and secondary crests as well as RAC, whereas the Lm was increased in *p21* knockout mice at both time points (Fig. 10d, e). Moreover, the number of vWF positive blood vessels in the lung was reduced in *p21* knockout mice compared to WT littermates at both time points (Fig. 10g).

We further administered the senolytic cocktail quercetin/dasatinib (2.5 mg/kg, i.p.) in WT mice at pnd1 and pnd3, and evaluated lung morphometry. As shown in Fig. 10h–j, administration of quercetin/dasatinib decreased numbers of senescent cells at pnd4, as indicated by reduced SA-β-gal activity, loss of nuclear lamin b1, and *p21* gene expression. This timing of injection led to decreased numbers of

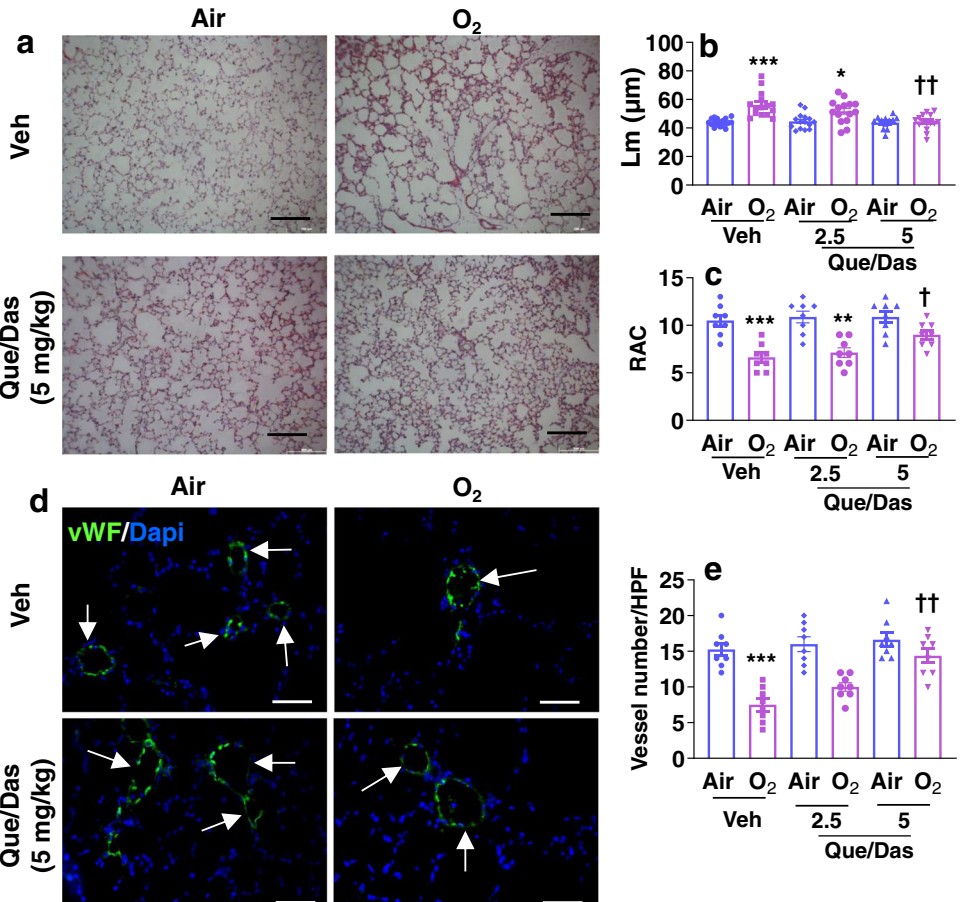

**Fig. 7 | Treatment with a senolytic cocktail during the alveolar stage protects against hyperoxia-induced alveolar and vascular simplification.** C57BL/6J neonatal mice (<12 h old) were exposed to air or hyperoxia (>95% $O_2$) for 3 days followed by air recovery until pnd7 or pnd14. Quercetin (Que, Q)/dasatinib (Das, D) (2.5 and 5 mg/kg) were intraperitoneally injected into mice at pnd4 and pnd6. **a** H&E staining was performed in the lung of hyperoxia-exposed mice treated with Que/Das (2.5 and 5 mg/kg) at pnd14. Bar size: 100 μm. **b**, **c** Lm and RAC were measured in these lungs. **d**, **e** Immunofluorescence of vWF was performed to count the number of vWF positive blood vessels in the lung. Arrows denote vWF positive blood vessels. Bar size: 50 μm. Data are expressed as mean ± SEM. *N* = 7 mice per group. Source data are provided as a Source Data file. One-way ANOVA followed by Tukey post-test was used for multiple comparisons. *$P < 0.05$, **$P < 0.01$, ***$P < 0.001$ vs corresponding Air group. †$P < 0.05$, ††$P < 0.01$ vs corresponding Veh/ $O_2$ group.

alveoli and secondary crests at both pnd7 and pnd14 (Fig. 10k, l). As expected, Lm was decreased and RAC were increased with advancing postnatal gestation in both quercetin/dasatinib-treated and control mice. However, pups treated with quercetin/dasatinib at pnd1 and pnd3 had a greater Lm and a lesser RAC at both pnd7 and pnd14 compared to control pups (Fig. 10m, n). The number of vWF positive blood vessels in the lung was reduced at both pnd7 and pnd14 after quercetin/dasatinib at pnd1 and pnd3 (Fig. 10o). Altogether, these results demonstrate that reducing senescent cells before and during the saccular stage disrupts lung development.

## Discussion

Mouse lungs at birth are structurally similar to human neonates born at 30 to 34 weeks of gestation, when the lung is in the saccular phase of development. Hyperoxic exposure in neonatal mice can be used to mimic the lung injury seen in premature infants with BPD. Allowing for air recovery tests whether the impact of hyperoxia is persistent, hence our model[19]. Here we show that lung cellular senescence is present at birth and reduces to adult levels after pnd3. Reducing numbers of lung cells programmed for senescence during this developmental period disrupts postnatal lung development. Interestingly, neonatal hyperoxia interrupts the normal reduction in developmental senescence by causing a transient increase during the alveolar stage. This leads to alveolar and vascular simplification.

These findings document the paradoxical importance of the timing of senescence in mediating saccular remodeling and also hyperoxia-induced alveolar simplification.

Cellular senescence contributes to embryonic development, which is mainly mediated by the p21 pathway[10–13,25]. This is corroborated by our findings that *p21* genetic deletion or pharmacological inhibition reduced developmental and hyperoxia-induced senescence, respectively. The phosphorylation event of p53 at S15 in humans and S18 in mice is able to increase its recruitment on the *p21* promoter, thereby augmenting *p21* gene expression[41,42]. Our recent report showed that *p53* deletion inhibits hyperoxia-induced senescence in cultured lung epithelial cells[15]. Thus, an activated phosphorylated p53/ p21 pathway contributes to developmental and hyperoxia-induced lung senescence. There were no changes in *p16* mRNA expression, another marker of senescence, in the lung regardless of hyperoxic exposure or different postnatal time points. This is corroborated by the fact that p16 is not associated with developmental senescence in embryonic kidney or limb[12,13]. Although phosphorylated p53 is associated with increased apoptosis in neonatal lungs exposed to hyperoxia[26], there was no colocalization of phosphorylated p53 with apoptotic biomarkers in hyperoxia-exposed lung at pnd7. Further studies using transgenic mice with a mutated S18 at p53[43] are warranted to better understand the role of phosphorylated p53 (ser 18) in regulating neonatal hyperoxia-induced senescence and apoptosis.

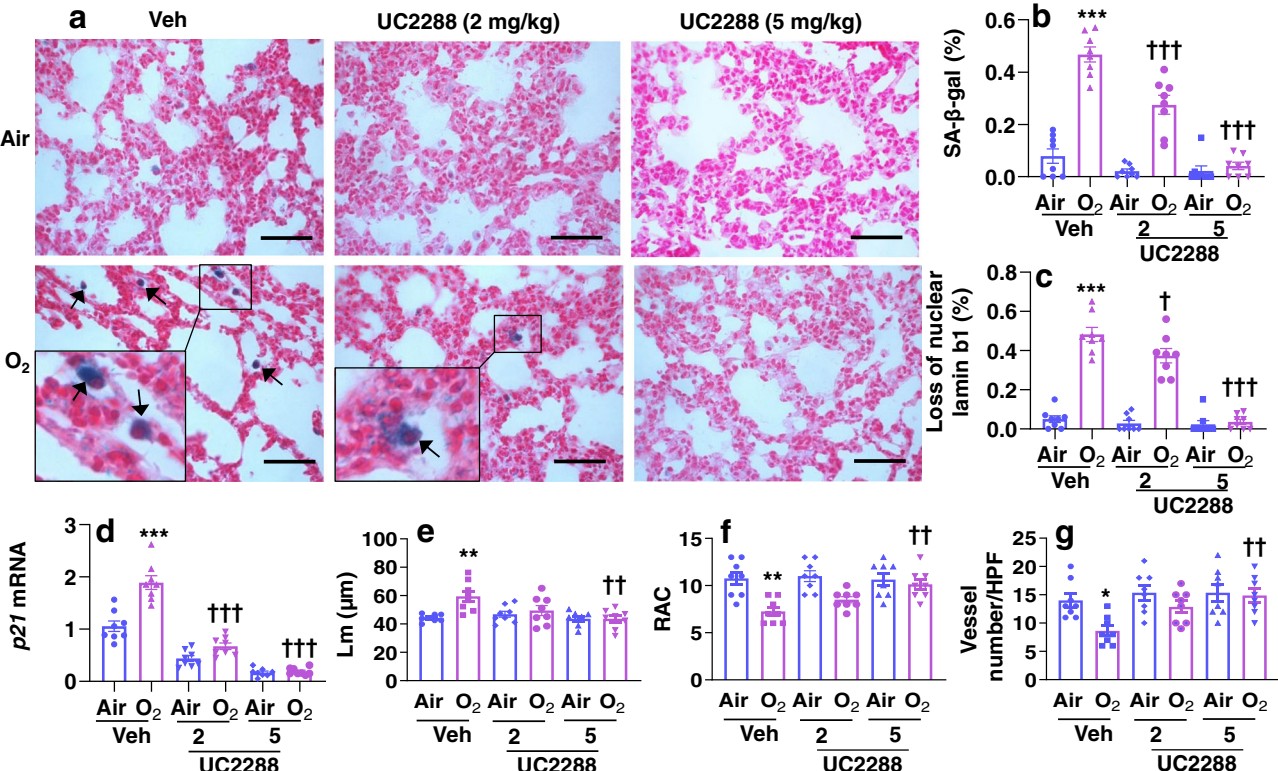

**Fig. 8 | UC2288 treatment during the alveolar stage reduces senescence makers and protects against hyperoxia-induced alveolar and vascular simplification.** C57BL/6 J neonatal mice (<12 h old) were exposed to air or hyperoxia (>95% $O_2$) for 3 days followed by air recovery until pnd7 or pnd14. UC2288 (2 and 5 mg/kg) were intraperitoneally injected into mice at pnd4 and pnd6. **a**–**c** SA-β-gal activity and loss of nuclear lamin b1 were assessed in the lung at pnd7. Bar size: 50 μm. **d** *p21* gene expression was measured by qRT-PCR in the lung at pnd7. **e**, **f** H&E staining was

performed in the lung of hyperoxia-exposed mice treated with UC2288 at pnd14. Lm and RAC were measured in these lungs. Bar size: 25 μm. **g** Immunofluorescence of vWF was performed to detect the number of vWF positive blood vessels in the lung. Data are expressed as mean ± SEM. *N* = 6–8 mice per group. Source data are provided as a Source Data file. One-way ANOVA followed by Tukey post-test was used for multiple comparisons. *$P < 0.05$, **$P < 0.01$, ***$P < 0.001$ vs corresponding Air group. ††$P < 0.01$, †††$P < 0.001$ vs corresponding Veh/$O_2$ group.

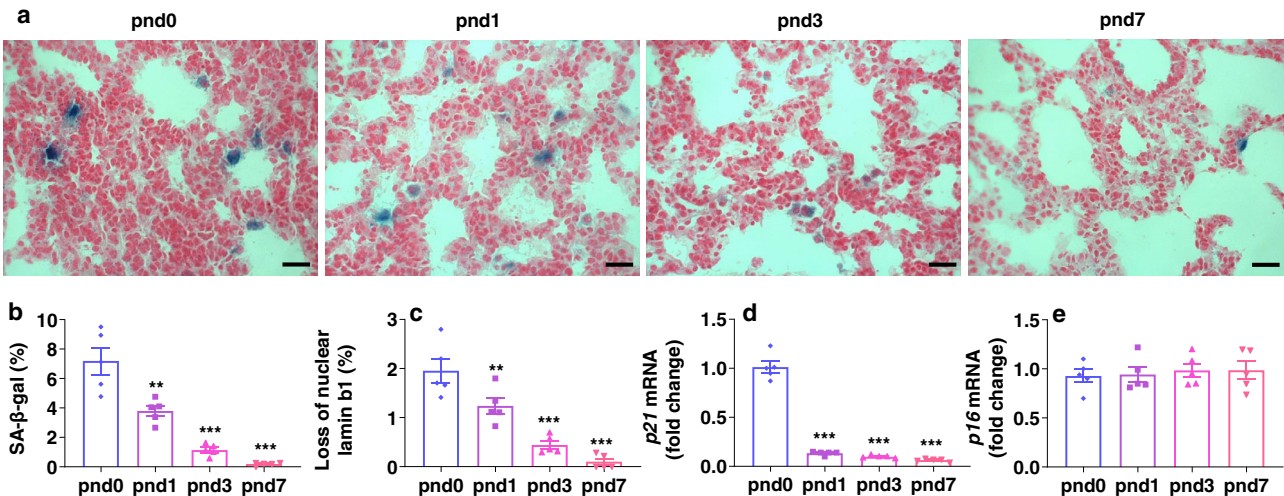

**Fig. 9 | Lung senescence occurs in newborn mice, and this gradually decreases during the saccular stage of lung development.** Lung was collected from C57BL/ 6 J mice at pnd0, pnd1, pnd3 and pnd7 under normoxia. **a**–**c** SA-β-gal activity and lamin b1 staining were performed in the lung. **d**, **e** qRT-PCR was carried out to

measure *p21* and *p16* gene expression in the lung. Bar size: 25 μm. Data are expressed as mean ± SEM. *N* = 6 mice per group. Source data are provided as a Source Data file. One-way ANOVA followed by Tukey post-test was used for multiple comparisons. **$P < 0.01$, ***$P < 0.001$ vs pnd0.

It is important to note that SA-β-gal activity can be induced in macrophages incubated with M2-inducing agents[44]. The number of lung M2 macrophages increases postnatally and peaks in the alveolar stage, and this is associated with alveolar formation[45]. This activity may contribute to the increased senescent cells detected by SA-β-gal

activity compared to those detected by loss of nuclear lamin b1 that we observed. Increased SA-β-gal activity may have also resulted in larger numbers of macrophages isolated by flow cytometric cell sorting based on $C_{12}$FDG staining. M2 macrophages usually have high phago-cytic capacity. We noticed that clusters 13 and 15 of macrophages

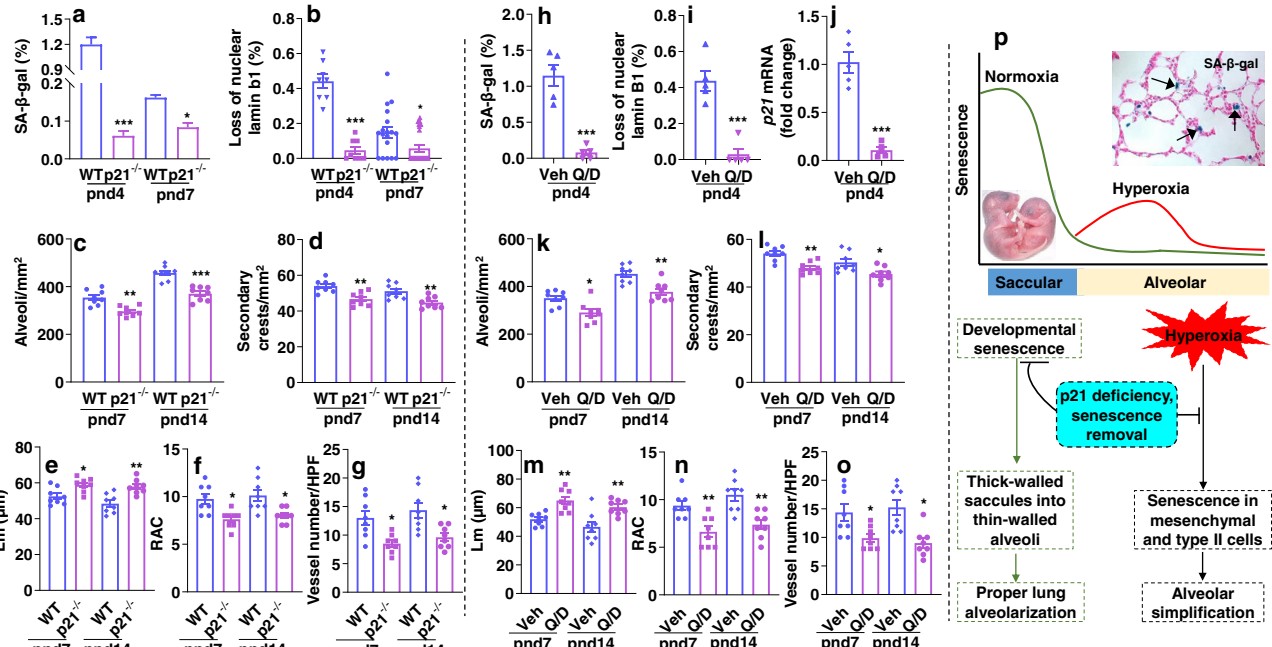

**Fig. 10 | Embryonic deletion of *p21* or treatment with a senolytic cocktail during the saccular stage disrupts lung development. a–f** Lungs from *p21* knockout mice and WT mice were collected at pnd4, pnd7 or pnd14 under normoxia. **a, b** Lung SA-β-gal activity and lamin b1 staining were performed, and cells positive for SA-β-gal or without nuclear lamin b1 were counted and normalized to total numbers of nuclei. **c–f** H&E staining was performed in the lung at pnd7 and pnd14 to measure number of alveoli and secondary crests, Lm and RAC. **g** Immunofluorescence of vWF was performed to detect the number of vWF positive blood vessels in the lung. **h–o** Lungs from C57BL/6 J mice were collected at pnd4, pnd7 or pnd14 under normoxia when they were treated Que/Das (2.5 mg/kg) at pnd1 and pnd3. **h–j** SA-β-gal activity, lamin b1 expression, and *p21* mRNA was

measured in the lung at pnd4. Cells positive for SA-β-gal or without nuclear lamin b1 were counted and normalized to total numbers of nuclei in the lungs. **k–n** H&E staining was performed in the lung at pnd7 and pnd14 to measure number of alveoli and secondary crests, Lm and RAC. **o** Immunofluorescence of vWF was performed to detect the number of vWF positive blood vessels in the lung at pnd7 and pnd14. **p** Schematic figure showing timing of senescence in mediating postnatal lung development and hyperoxic lung injury. Data are expressed as mean ± SEM. *N* = 6–8 mice per group. Source data are provided as a Source Data file. One-way ANOVA followed by Tukey post-test was used for multiple comparisons (**a–g**, **k–o**), while *t*-test was used in panels (**h–j**). *$P < 0.05$, **$P < 0.01$, ***$P < 0.001$ vs corresponding WT (**a–g**) or vehicle (**h–o**) group.

expressed both M2 markers and phagocytosis regulatory genes, suggesting that these subpopulations are polarized into an M2 status for phagocytosis. Interestingly, senescent macrophages can be polarized into M2 phenotype, and activate a pro-fibrotic transcriptional program[34,44]. It is unknown whether these subpopulations also play important roles in mediating lung fibrogenesis in hyperoxia-exposed mice. Further studies are warranted to understand whether the influx of lung macrophages seen in hyperoxia protects the lung by engulfing senescent, apoptotic or necrotic lung cells or are senescent themselves and increase lung injury and fibrosis.

Immunostaining of autopsy samples shows increased triple localization of p21, γH2AX and smooth muscle actin in the airways of infants exposed to 4 days of hyperoxia compared with those who died within 1 day of birth[14]. This suggests that the DDR participates in lung cellular senescence in neonatal hyperoxic injury. Indeed, both γH2AX and 53BP1 are part of the early cellular signaling of DNA injury, and initiate the repair of DNA double-strand breaks. Nevertheless, hyperoxia-exposed neonatal rats have increased lung 8-oxo-DG, which is associated with apoptosis[46]. It is therefore not completely understood whether increased DNA damage results in hyperoxia-induced apoptosis and/or senescence in different cells. No changes in γH2AX or 53BP1 positive cells were observed in the lung during developmentally regulated senescence. This is corroborated by the previous report showing that DNA damage is not associated with developmental senescence in other tissues[13].

Cellular senescence results in proliferation arrest. Although normal cellular proliferation peaks early in postnatal lung development, the further decline observed with hyperoxic exposure agrees with previous observations[47]. Whether this reduced proliferation is due to the

contribution of non-proliferative senescent cells is unclear. Hyperoxia-induced reduction in proliferation at the saccular and alveolar stage could also be due to increased p21 expression[48]. A previous study of mouse forelimb formation demonstrated that a subset of senescent cells lost their senescence hallmarks, and these cells can re-enter the cell cycle and resume proliferation[25]. When the neonatal mice were recovered in air until adulthood, there were no differences in lung cell proliferation between the air and hyperoxia groups. This suggests that the cells with reduced proliferation may have restored their ability to proliferate in the alveolar stage or that there was compensatory proliferation of other cells. Lineage tracing experiments demonstrate that neonatal hyperoxic exposure stimulates type II cell proliferation in the saccular stage[47]. Indeed, a cluster of type II cells highly expressing Malat1, a non-coding RNA gene responsible for proliferation, was highly increased by neonatal hyperoxia[19]. Further studies using lineage tracing experiments would elucidate whether neonatal hyperoxia causes dynamic changes in senescence and proliferation in specific lung cell types during postnatal lung development.

The pulmonary mesenchyme directs lung branching morphogenesis, epithelial differentiation, lineage distinction, vascular development, and alveolar maturation. The interaction between the mesenchyme and the epithelium is essential for lung development and repair[49,50]. We identified that both Pdgfra⁺ secondary crest myofibroblasts and type II cells were senescent after hyperoxic exposure. Type II epithelial cells are precursors of type I cells and can generate vascular mitogens, which are disrupted by neonatal hyperoxia[51]. Therefore, neonatal hyperoxia-induced senescence in these type II progenitor cells may result in lung injury. Sox9 is a stem cell associated transcription factor which plays multiple roles in the lung epithelium during

branching morphogenesis[38]. Although Sox9 levels are drastically reduced in mouse lung after birth[37], neonatal hyperoxia increases lung Sox9 expression in the alveolar stage. Increased Sox9 expression after hyperoxia may represent a compensatory response whereby developmental pathways are reactivated in an attempt to repair the lung[52]. Interestingly, neonatal hyperoxia also caused senescence in Sox9 positive cells but reduced *Sox9* gene expression in senescent type II cells. This would result in diminution of the compensatory effect of Sox9 by preventing the proliferation of progenitor cells through senescence, thereby leading to insufficient repair of the injured lung after hyperoxia. This hypothesis is corroborated by the fact that reducing senescent cells after hyperoxia protected against hyperoxia-induced alveolar simplification. During postnatal development, we did not observe senescence in type II cells or secondary crest myofibroblasts in the lung. Further study is warranted to determine whether senescence in vimentin positive cells contributes to thinning of the lung mesenchymal compartment to accommodate the developing airways and air sacs needed for gas exchange during lung development.

Senescent cells usually release SASP, which reinforces and spreads senescence in host tissues[53,54]. In the present study, neonatal hyperoxia increased SASP factors in senescent cells. This agrees with the in vitro findings that hyperoxic exposure increased levels of PAI-1, IL1-α, IL1-β, IL-6, LAP, and TNF-α in cultured fibroblasts[16]. Although the overall number of senescent cells is small (0.3% of total lung cells) in the lung of mice exposed to hyperoxia, SASP factors secreted from senescent cells are able to exert a paracrine effect and subsequent lung injury by propagating the senescence program as well as inducing inflammation or apoptosis. In contrast, blocking IL-1 or Cxcl2 receptors prevents hyperoxic lung injury[55,56]. Additionally, inhibiting inflammatory responses by quercetin alone (20 mg/kg, i.p.) protects against neonatal hyperoxia-induced alveolar simplification in mice[57]. This is similar to what has been shown previously that transplanting a small number of senescent preadipocytes (1 million cells) into the peritoneal cavity was sufficient to induce physical dysfunction, including reduced maximal walking speed, grip strength and hanging endurance, in young adult mice[54]. Different SASP factor profiles were observed in developmentally regulated senescence compared to hyperoxia-induced senescence. A decrease in TNF-α, MMP-8 and PAI-1 levels paralleled with reduced senescent cells during postnatal lung development, suggesting these factors are released from these senescent cells. Tnfsf12 (Tweak) is a mitogenic and migration factor for endothelial cells which also promotes angiogenesis[58]. Increased extracellular matrix degradation by MMP-12 may further create a microenvironment which allows endothelial cell migration and proliferation induced by Tnfsf12 for vascularization during lung development. Additionally, reducing senescent cells during the saccular stage pharmacologically disrupts lung development, as evidenced in *p21* knockout mice in this study and a previous report[59]. This may further increase the susceptibility to hyperoxia-induced lung injury[59]. These findings demonstrate that developmentally programmed senescence is essential for promoting lung tissue remodeling and development. Given the role of senescent cells in lung development and hyperoxic lung injury, the development of novel senolytics that do not clear developmentally regulated (beneficial) senescence is warranted for treating hyperoxic lung injury, as is careful attention to the timing of use of senolytics to obviate elimination of developmentally regulated senescence.

In conclusion, lung senescence is normally observed in vimentin[+] mesenchymal cells in the saccular stages of the newborn mouse lung, and this is essential for lung development. Neonatal hyperoxia increases cellular senescence markers, particularly in type II cells, Pdgfra[+] mesenchymal and immune cells, during the alveolar stage. This process is mediated by the p53/p21 pathway. A hyperoxia-induced senescent phenotype results in alveolar and vascular simplification (Fig. 10p). Our novel findings provide mechanisms underlying postnatal lung development and help guide optimal timing for therapeutic interventions to mitigate neonatal lung injury by reducing senescent cells in the alveolar stage while preserving developmental senescence in the saccular stage.

## Methods

All animal experiments were approved by the Institutional Animal Care and Use Committee of Brown University (IACUC#: 21-08-0003). Utilization of post-mortem human lung samples was approved by the Institutional Review Board (IRB#: 792344-9) of Women and Infants Hospital, Providence RI. Informed consent was obtained from the guardians of the deceased.

### Animals and hyperoxic exposure

Mice were maintained at room temperature and 55-60% humidity on 12 h light/dark cycles with *ad libitum* access to water. Newborn C57BL/6 J (<12 h old) along with their mothers were exposed to room air or hyperoxia (>95% $O_2$) for 3 days in an A-chamber (BioSpherix, Redfield, NY)[60,61]. The dams were switched between exposed litters and room air control litters every 24 h in 3 days exposed animals to avoid maternal injury. For time points beyond pnd3, the pups were allowed to recover in room air until time of harvest. Constitutive p21 knockout mice (C57BL/6 background) were purchased from the Jackson (#016565), and housed in room air until time of lung harvest. Mice were monitored daily until sacrifice. At endpoints, mice were euthanized with an intraperitoneal injection of an overdose of ketamine (75 mg/kg) and xylazine (10 mg/kg).

### Drug administration

A combination of quercetin and dasatinib (2.5 mg/kg) was administered through intraperitoneal injection (i.p.) to mice at pnd1 and pnd3 during the saccular stage under normoxic condition. We also injected quercetin/dasatinib (2.5 and 5 mg/kg, i.p.) to mice at pnd4 and pnd6 during the alveolar stage after they were exposed to room air or hyperoxia for 3 days as neonates. Doses of quercetin and dasatinib was chosen based on previous studies showing that they removed senescent cells without developing toxicity[54]. Similarly, we administered a specific p21 inhibitor (UC2288, 2 and 5 mg/kg, i.p., Sigma) to mice at pnd4 and pnd6 after they were exposed to room air or hyperoxia for 3 days as neonates[62]. At pnd4, pnd7 and pnd14, mice were sacrificed and lungs were used for further analysis.

### Lung tissues from premature infants

Human lung samples were obtained from premature infants between 23 and 29 weeks postmenstrual age, who lived 5-15 days and required mechanical ventilation, and controls were premature infants who were not mechanically ventilated and survived less than 24 h[5]. Clinical characteristics of these subjects are shown in Supplementary Table 1.

### SA-β-gal activity assay

Fresh mouse lung samples were embedded in cryosectioning medium (Tissue-Tek OCT) and sliced at 8 μm thickness using a Leica cryostat[63]. Sections were placed on charged slides and kept on dry ice until staining. Lung sections were fixed for 15 min in 0.5% glutaraldehyde, then placed for 16 h in 1 mg/mL X-gal solution (pH 6.0) at 37 °C. Sections were then counter-stained with nuclear Fast Red (Sigma), dehydrated, and mounted in a xylene-based medium. Senescent cells (blue) were counted in a blinded manner and normalized into total nuclei (red) per high-power field in 3–5 images per animal at 400× magnification. The data are expressed as the percentage of SA-β-gal positive cells.

### Immunofluorescence

Lung tissues were embedded and sectioned at 5 μm thickness. Sections were de-paraffinized and subjected to heat-mediated antigen retrieval in a citrate buffer solution (Vector Labs), then incubated with

antibodies against lamin b1, p21, Ki67, γ-H2AX, 53BP1, 8-oxo-DG, phosphor-p53, Hopx, vWF, vimentin, pro-SPC, Pdgfra, Sox9, and F4/80 overnight at 4 °C. Antibody information is shown in Supplementary Table 2. Samples were incubated with secondary antibodies for 1 h at room temperature, and then mounted in hard-set mounting medium containing DAPI (Vector Labs). Tyramide signal amplification was employed for dual immunofluorescence using two primary antibodies raised in the same host species. Images were taken using an AxioVision software (v4.8.2) under a Zeiss Axiovert 200 M Fluorescence Microscope. Cells expressing the markers of interest were counted and normalized into total nuclei in 3–5 images per animal at 400× magnification in a blinded manner.

### Quantification of DNA damage foci and DNA oxidation

After immunofluorescence, an ImageJ macro (v1.53t) was used to quantify percentages of cells containing 53BP1, γH2AX, or 8-oxo-DG foci[64]. A threshold is set for what is included in the image for the antibody being studied, so as to not include any background fluorescence in the analysis. The macro then scans the DAPI image. It sets a threshold for nuclear size, not including anything too small to be a nucleus or so large that it might include more than one nucleus. Both images are then overlayed on each other, and the macro scans for co-localization. The output is the number of nuclei containing foci relative to the total number of nuclei in the image.

### Western blot

Cells and lung tissues were homogenized in a tissue grinder with radioimmunoprecipitation assay (RIPA) buffer. Protein levels were measured using a Pierce BCA Protein Assay kit (Thermo Scientific, Rockford, IL, USA). Protein samples (5–15 µg) from lysates were separated on a NuPAGE 4–12% Bis-Tris protein gel (Invitrogen), and separated proteins were electroblotted onto nitrocellulose membranes. The membranes were blocked for 1 h at room temperature with 5% milk, and then probed with 1:1000–1:10000 diluted antibodies to determine the corresponding protein levels. Antibodies used are shown in Supplementary Table 2. After three washing steps (5 min each), the levels of protein were detected using secondary antibodies (1: 5000 dilution in 5% milk in PBS containing 0.1% Tween-20 for 1 h) linked to horseradish peroxidase (Vector Laboratories). The bound complexes were detected by the ChemiDoc Touch Imaging System (BIO-RAD) using the enhanced chemiluminescence method (Millipore). Equal loading of the samples was determined by quantification of proteins as well as by reprobing membranes for the housekeeping control calnexin.

### Measuring phosphorylation of p53

Lung tissue was homogenized or cells were lysed in RIPA buffer containing phosphatase inhibitors. Samples were loaded in duplicate onto a plate, and phosphorylation of p53 was measured using the phosphorylated Ser15 p53 ELISA kit (Cell Signaling #7365) according to manufacturer's instructions[15].

### RNA extraction and real-time PCR

Tissue samples were homogenized in a tissue grinder using TRIzol reagent and purified using the RNeasy miniprep kit (Qiagen, Valencia, CA)[60]. RNA samples were quantified with the NanoDrop One Micro-volume UV-Vis Spectrophotometer (Thermo Fisher Scientific, Wilmington, DE). Then, 400 nanograms of total RNA in a volume of 20 µl was used for reverse transcription using Taqman Reverse Transcription Reagents (Thermo Fisher Scientific). One microliter of cDNA was used for real-time PCR reactions in the 7300 Real-Time PCR System (Applied Biosystems). All Taqman gene probes were purchased from Thermo Fisher Scientific (see Supplementary Table 2). Gene expression was normalized to 18 s rRNA levels. Relative RNA abundance was quantified by the comparative $2^{-\Delta\Delta Ct}$ method.

### Proliferation assay

For the lung proliferation assay, mice were intraperitoneally injected with EdU (50 mg/kg) once for mice sacrificed at pnd3, pnd7 and pnd10 or for 3 consecutive days in adult mice, and then sacrificed 24 h after the last EdU injection. The Click-iT EdU Cell Proliferation Kit for Imaging, Alexa Fluor™ 647 (Invitrogen) was used to determine proliferation in the lung[61]. Proliferation was quantified by counting EdU-positive cells and total nuclei per field in 3–5 images per animal at 400× magnification using a Zeiss Axiovert 200 M Fluorescence Microscope.

### Luminex assay

Cytokine levels in lung homogenates and cell culture supernatants were measured using a custom R&D systems Murine Premixed Multi-Analyte Kit (R&D Systems, Inc., Minneapolis, MN) and a Luminex 200 Instrument (Luminex Corporation, Austin, TX) according to the manufacturer's instructions. Samples were stored at −80 °C until the day of analysis when they were then thawed and centrifuged to pellet debris. Levels of TNF-α, IL-1α, IL-1β, Cxcl2/MIP-2, IL-10, Serpin E1/PAI-1, TNFSF12/Tweak, Cxcl12, MMP-8, MMP-12, and TNFRII/TNFRSF1B were measured. Samples were run in duplicate and analyte concentrations were reported in picograms per milliliter (pg/ml). Cytokine concentrations were calculated using a six-point standard curve derived from measurement of serially-diluted standards run in duplicate. Upper and lower limits of detection were individually determined for each analyte, based on the included standards.

### scRNA-seq

At pnd7, lung from hyperoxia-exposed mice was disassociated into single cell, and suspensions from three mouse lungs were pooled as shown in our recent report[19]. The FACS was utilized to sort $C_{12}$FDG positive cells from the lung suspensions. Single cell encapsulation was performed using the Chromium Single Cell Chip G kit on the 10× Genomics Chromium Controller, and single cell cDNA was sequenced by Genewiz on the Illumina Hiseq (2 × 150 bp paired-end runs). We also revisited and mined our published scRNA-seq dataset from hyperoxia- or air-exposed mice at pnd7 and pnd60 as controls[19]. In these datasets, we did not specifically sort $C_{12}$FDG positive cells. Quality control of the data was performed using the R package Seurat v 3.2.1[19]. Briefly, data were filtered to remove doublets and dead cells by retaining cells with between 700 and 8000 genes (nfeatures) and less than 5% of reads of mitochondrial origin. Datasets were integrated using Seurat SelectIntegrationFeatures with nfeatures of 3000 and clustering was performed on 50 dimensions and a resolution of 2. Cell states were identified using Seurat TransferData functions with the Mouse Cell Atlas data as the reference group.

To obtain less granular cell state annotations, the Mouse Cell Atlas cell types were further binned into groupings as follows: Type I and type II cells were left as their own distinct groupings; Ciliated cell', 'Dividing cells', and 'Club Cell' were assigned to 'Other epithelial cells'; 'Alveolar macrophage_Ear2 high', 'Alveolar macrophage_Pclaf high', and 'Interstitial macrophage' were assigned to 'Macrophages'; "Stromal cell_Acta2 high', 'Stromal cell_Inmt high', and 'Stromal cell_Dcn high' were assigned to 'Mesenchymal cells'; 'Endothelial cell_Tmem100 high', 'Endothelial cell_Kdr high', and 'Endothelial cells_Vwf high' were assigned to 'Endothelial cells'; and 'Basophil', 'T Cell_Cd8b1 high', 'B Cell', 'Nuocyte', 'Conventional dendritic cell_Gngt2 high', 'Eosinophil granulocyte', 'Plasmacytoid dendritic cell', 'Dendritic cell_Naaa high', 'Dividing T cells', 'NK Cell', 'Conventional dendritic cell_H2-M2 high', 'Neutrophil granulocyte', 'Conventional dendritic cell_Mgl2 high', 'Conventional dendritic cell_Tubb5 high', 'Dividing dendritic cells', and 'Ig-producing B cell' were assigned to 'Other immune cells'. SCTransform was used for normalization of all datasets.

Marker genes for the $C_{12}$FDG positive (senescent) cells were identified by comparing them to the pnd7/air and pnd7/hyperoxia group cells without $C_{12}$FDG sorting using Wilcoxon rank sum test, as

implemented in Seurat's FindMarkers function, which makes pairwise comparisons between specific identity classes. In this comparison, the identity classes were the samples originated from pnd7/hyperoxia group $C_{12}FDG$ positive cells vs pnd7/hyperoxia group cells without $C_{12}FDG$ sorting (Supplementary Data 2), pnd7/$C_{12}FDG$ positive cells vs pnd7/air group cells without $C_{12}FDG$ sorting (Supplementary Data 7), and pnd7/hyperoxia group cells without $C_{12}FDG$ sorting vs pnd7/air group cells without $C_{12}FDG$ sorting (Supplementary Data 8). Marker genes for the cell states only within the pnd7 $C_{12}FDG$ positive (senescent) cells were identified by using the Wilcoxon rank sum test, as implemented in Seurat's FindAllMarkers function. This function compared each individual identity class to all other identities in the object. In this comparison, the identity classes were the Seurat clusters (Supplementary Data 9). The SCT assay was used for all of the tests.

Marker genes were binned as being type II marker genes based on our previous study[19], involved in phagocytosis if they were in 'R-MMU-2029480' (Fcgamma receptor dependent phagocytosis), 'R-MMU-2029485' (Role of phospholipids in phagocytosis, 'R-MMU-2029482' (Regulation of actin dynamics for phagocytic cup formation), or 'R-MMU-8941413' (Events associated with phagocytolytic activity of PMN cells) reactome pathways as per BioMaRt[65,66]. All of the code is available at https://zenodo.org/record/7401907.

### Senescent type II cell isolation and culture

Lung type II cell isolation was done by following the methods from Corti et al.[67] with minor modifications. At pnd7, lungs collected from mice exposed to room air or hyperoxia as neonates were minced and digested with dispase II (Sigma) and DNase I (Sigma). Red blood cells were removed from single cell suspensions using the RBC lysis buffer (Invitrogen). After the depletion of leukocytes with CD45 microbeads (Miltenyi Biotec), type II epithelial cells were positively selected using a FITC anti-mouse SPC antibody (Santa Cruz) and anti-FITC microbeads (Miltenyi Biotec) using the BD FACSAria (BD Bioscience). The purity of isolated cells was assessed by immunofluorescence. The second cell sorting was performed to isolate $C_{12}FDG$ positive cells for following culture. Cells were cultured in a 6-well plate with approximately $2 \times 10^6$ cells per well with DMEM/F-12 medium containing 10% FBS, glutamine and antibiotics for 24 h. Cells were stained for pro-SPC, while supernatants were collected for further analysis and administration.

### Administration of supernatants from senescence type II cells into neonatal mice

At pnd7 and pnd10, 20 µl of above culture supernatants with or without a 1:1 dilution (1 part supernatant to 1 part saline) were administered intranasally into C57BL/6 J neonatal mice under isoflurane anesthesia. Lung morphometry including Lm and RAC was measured at pnd14.

### Lung morphometry

Non-lavaged mouse lungs were inflated with 1% low melt agarose at a pressure of 20 cm $H_2O$, and fixed them with 4% neutral buffered paraformaldehyde[68]. These fixed lungs were embedded in paraffin and sectioned into 5 µm sections using a rotary microtome (MICROM International GmbH), then stained with hematoxylin and eosin. Total alveolar numbers were calculated from the lung midsagittal sections to determine Lm of airspace using the MetaMorphTM software (Molecular Devices). Briefly, ten randomly selected ×10 fields per slide were photographed in a blinded manner, and the images were manually thresholded. The airway and vascular structures were eliminated from the analysis. Respiratory bronchioles partly lined by epithelium were selected. A perpendicular line was drawn from the center of the respiratory bronchiole to the distal acinus (as defined by the pleura or the nearest connective tissue septum). The number of septae intersected by each line was counted as RAC, and a minimum of 10 counts were performed for each animal[69]. Slides were stained for elastin to enhance recognition of secondary crests[70]. Secondary crest volume

density was measured using a 130-point contiguous counting grid superimposed on each (×400) image. The data are expressed as the number of secondary crests per field.

### Statistics

Statistical analyses were performed using GraphPad Prism 9.4.0. The results were expressed as mean ± SEM. The $t$-test was used for detecting statistical significance of the differences between means of two groups after checking the normality of data. The statistical significance of the differences among groups was evaluated by using one-way ANOVA for overall significance, followed by Tukey's multiple comparisons test. Statistical significance was considered existing when $P < 0.05$.

### Reporting summary

Further information on research design is available in the Nature Portfolio Reporting Summary linked to this article.

## Data availability

The scRNA-seq generated in this study (Mouse Cell Atlas data) have been deposited in the Gene Expression Omnibus database under primary accession code GSE207866. All other data needed to evaluate the conclusions of the study are present in the paper or in the Supplementary files. Source data are provided with this paper.

## Code availability

All of the Seurat code and the Docker file for the environment is available on Zenodo at https://zenodo.org/record/7401907.

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

## Acknowledgements

This work was supported by the Brown University Research Seed Award, the Warren Alpert Foundation (to P.A.D.), an Institutional Development Award (IDeA) from the NIGMS of NIH under grant #P20GM103652, the Dr. Ralph and Marian Falk Medical Research Trust Bank of America, N.A., Trustee (to H.Y.). Part of this research was supported by an IDeA from the NIGMS of NIH under grant number P20GM109035 (J.W.). The funders had no role in study design, data collection and analysis, decision to publish, or preparation of the manuscript. Some of these results were presented at the American Thoracic Society International Conference, May, 2019 and 2022, and published in abstract form in American Journal of Respiratory and Critical Care Medicine.

## Author contributions

H.Y. and P.A.D. conceived and designed experiments. H.Y., J.W., A.L.P., A.S., S.R., K.H., H.M., J.C., N.O., J.A.K., K.E.H., M.D.P. and G.B. acquired and analyzed the data. H.Y. and P.A.D. interpreted the data. H.Y. drafted the manuscript. H.Y. and P.A.D. revised the manuscript.

## Competing interests
The authors declare no competing interests.
