## [Peer Review File · Nature Communications]

Reviewer comments, first round

Reviewer #1 (Remarks to the Author):

Specific Comments:

1. What are the mechanisms that contribute to the inhibition of oxygen induced simplification caused by deletion of senescent cells?
2. Many of the senescent cells sorted with C12FDG are inflammatory cells; do these cells contribute to SASP and alveolar remodeling?
3. Do the senescent AT2 cells contribute to SASP?
4. SOX9 staining is diffuse; are these cells epithelial or mesenchymal - fibroblasts? SOX9 expression in the epithelium is normally strongly repressed in the alveolar period of morphogenesis. Colocalization with established cell-specific markers are needed. Do senescent AT2 cells or AT1 cells contribute to the simplification? Some PDGFRA mesenchymal cells are known to play a critical role in alveologenesis - are these senescent cells lost during O₂ induced simplification.
5. Can the authors comment on the mechanisms by which programmed senescence promotes normal alveologenesis?

Figure 1C – the quality of the images is distinct; O₂ exposed image has considerable background.

Figure 1E, F – The history, diagnosis, gestational and postnatal ages of the patients included in the analysis should be provided. Supplemental data showing histology of these lung sections are needed to understand regional pathology.

Suggest moving Figure 2 to the Supplement or combining panels rather than multiple separate figures.

Figure 4F – The analysis, algorithms used, data quality, numbers of cells, and statistical analysis are needed to support the figure. Raw data should be made available, or the accession number should be provided if published. If data are from previously published and analyzed, this might be stated and referenced in the legend to clarify its origin. The specific cell types contributing to SASP (4F) should be indicated and statistically validated.

Figure 5 – Are AT1 cells also senescent? Which fibroblastic cell types?

Figure 6 – SOX9 staining is diffuse. It is usually nuclear; there is high background. What is the cell type of the SOX9+ cells?

Reviewer #2 (Remarks to the Author):

General

This manuscript describes the detection of senescence during lung development and in response to hyperoxia-induced injury. Data on using the senolytic cocktail (quercetin and dasatinib) during the sacular and alveolar stage of lung development is provided. In addition, the investigators have used the p21 null mutant neonatal mouse and a selective p21 inhibitor to study the role of senescence in postnatal lung development and after hyperoxia exposure.

Specific

Major Concerns:

1. Results. Fig.1F. Were the investigators not able to detect SA- β -gal in the human premature infants?
2. Results. Fig. 1H. Based on Edu staining only, the authors give a strong statement that lung senescence is reduced in the sacular stage. To prove this both Ki67 and Edu should be done because Ki67 detects all stages of cell division (except G₀) while Edu labels only during the "S" phase.

3. Results. Fig. 2. The authors have shown there is no sex differences in senescence, but only at 1 time point (pnd7). Other time points should have been shown for consistency and confirmation. This is important for translational significance in this experimental BPD mouse model, as it is well known that there is a preponderance of male infants in the development and severity of human BPD (for e.g. see PMIDs: 31727986; 24421662).

4. Results. Fig. 3F. Interestingly, other authors have shown that p53 protein is increased in neonatal mouse lungs after hyperoxia on pnd3. See Fig. 6 in PMID: 21270677. How do the investigators explain their result of no difference in p53 protein results? Furthermore, the investigators show phospho-p53 at S18 in the mouse lung is increased and claim this suggests cellular senescence is increased. However, phosphorylation of p53 is well described to be increased in apoptosis in neonatal lungs exposed to hyperoxia (for e.g. see PMID: 11435201). Thus, the authors shown an association, not causation, as regards senescence.

5. Results. Fig. 4D. 8-oxo-DG has been shown to be increased in hyperoxia-exposed neonatal rats in association with apoptosis (see PMID: 32933455).

6. Results. Fig. 4F. Senescence leading to inflammation and the presence of inflammatory markers have not been confirmed by protein data.

7. The selective p21 inhibitor (UC2288) is also known to attenuate apoptosis, and "alternative method by which p21 might protect against hyperoxia is through its ability to block apoptosis" (PMID: 16723699). How do the authors reconcile the blocking of the apoptosis pathway (versus senescence) in terms of improving the lung morphometry?

8. As also mentioned by the authors (ref.# 45), p21 null mutant neonatal mice have been described in a 2004 report to have worse lung morphometry upon hyperoxia-exposure.

9. Incidentally, quercetin alone has been used in neonatal mice exposed to hyperoxia and led to attenuation of hyperoxia-induced lung injury. Please see PMID: 29432836.

10. Surprisingly, there is no description of the impact of the above-mentioned manipulations on the vascular phenotype in the experimental BPD mouse model.

Reviewer #3 (Remarks to the Author):

The manuscript by Yao et al (NCOMMS-21-42376) describes the identification of developmentally-regulated senescent cells in the mouse lung at birth that decrease throughout the saccular stage. In addition, exposure of newborn mice to hyperoxia at pnd3 led to increased senescence, mediated apparently by p53/p21 in mesenchymal and type II cells. Eliminating these O₂-induced senescent cells with the senolytic cocktail of dasatinib + quercetin (D+Q) during the alveolar stage inhibited hyperoxia-induced alveolar simplification. The authors conclude programmed senescence orchestrates postnatal lung development whereas later hyperoxia-induced senescence causes alveolar simplification. These results are of potential interest since they suggest that senolytic treatment during the alveolar stage could be used therapeutically to prevent hyperoxia-induced alveolar simplification. However, much of the analysis is based on immunostaining, supported by scRNA seq in only one type of experiment. The authors need to provide further evidence that at least two markers of senescence (e.g., p21+, Lamin B-, HBGB1-, p53BP1 foci, TAFs, etc) co-localized to the same cell. Also, the authors need to characterize the developmentally regulated SnC at pnd0 in further detail in regard to their senescent signature compared to hyperoxia-induced senescence.

Specific Comments:

1. The authors need to co-localize their senescent cell markers to the same cell. For example, are

p21+ cells always Lamin B or HMGB1 low and p53BP1 positive?

2. It is somewhat surprising that qPCR and bulk RNAseq analysis did not reveal an increase in SASP factors. Although this could be due to low number of senescent cells, p21 is highly upregulated at the expression level. Does this mean the SnCs in the lung only express a low level of SASP?

3. It is hard to visualize in the images provided what the authors are calling positive or negative for immunofluorescence staining. Larger images should be included as supplemental figures.

4. The analysis of p53 phosphorylation by Elisa, especially with the small increases observed, isn't convincing. Can Western analysis of p53 phosphorylation in the C12FDG positive cells be performed? Alternatively, p21 and phospho-p53 can be co-localized by CyTOF.

5. The p21 inhibitor UC2288 supposedly decreases p21 mRNA expression independently of p53. Does this mean that the upregulation of p21 in senescent cells is independent of p53?

6. Is there disrupted lung development in p53 KO mice?

7. The authors attempt to characterize the hyperoxia induced senescent cells, but not the ones elevated at pnd0. Do these developmentally regulated senescent cells express a similar SASP as the hyperoxia induced senescent cells? Do they express markers of DNA damage? What type of cells are senescent? Do they also express Sox9?

8. There are preliminary reports of the senolytic combination of D+Q not killing certain "beneficial" senescent cells so the fact that these cells appear to be cleared by D+Q is of interest. It is also of interest since clinical trials to treat eclampsia are being planned. Thus these developmentally-regulated SnCs need to be characterized further.

Manuscript #: NCOMMS-21-42376

Timing and cell specificity of senescence drives postnatal lung development and injury

Yao et al.

Reviewer 1's comments:

Point 1: What are the mechanisms that contribute to the inhibition of oxygen induced simplification caused by deletion of senescent cells?

Response: We thank the reviewer for this thoughtful comment. In the revised manuscript, we demonstrate that the SASP factors (e.g., IL-1 α , IL-1 β , and Cxcl2) secreted from senescent cells cause lung injury by injecting these into mouse lungs and observing deleterious effects consistent with the damage from hyperoxia. In addition, blocking IL-1/IL-1R or Cxcl2/Cxcr2 signals prevents alveolar simplification in response to hyperoxic exposure in mice by reducing the activation of macrophages and neutrophils (PMID: 23946428, 15004193). Therefore, we conclude that reduced SASP factors contribute to the inhibition of oxygen-induced simplification caused by deletion of senescent cells.

Point 2. Many of the senescent cells sorted with C12FDG are inflammatory cells; do these cells contribute to SASP and alveolar remodeling?

Response: As shown in **Supplementary Figure 4a**, these inflammatory cells also expressed high levels of SASP genes including Cxcl2, Tnfaip1, and IL-1 β . A previous report showed that IL-1 receptor antagonist prevents neonatal hyperoxia-induced alveolar simplification (PMID: 23946428). Thus, these senescent inflammatory cells contribute to hyperoxia-induced SASP and possible subsequent alveolar simplification.

Point 3. Do the senescent AT2 cells contribute to SASP?

Response: According to **Supplementary Figure 4a**, AT2 cells expressed high levels of SASP genes. Furthermore, protein levels of SASP factors (e.g., IL-1 α , IL-1 β , Cxcl2 and TNFRSF1B) were also increased in supernatants of senescent AT2 cells isolated from hyperoxia-exposed mice (**Figure 6**). Thus, the senescent AT2 cells contribute to SASP.

Point 4. *SOX9 staining is diffuse; are these cells epithelial or mesenchymal - fibroblasts? SOX9 expression in the epithelium is normally strongly repressed in the alveolar period of morphogenesis. Colocalization with established cell-specific markers are needed. Do senescent AT2 cells or AT1 cells contribute to the simplification? Some PDGFRA mesenchymal cells are known to play a critical role in alveologenesi s - are these senescent cells lost during O₂ induced simplification.*

Response: As suggested, we have reperformed SOX9 staining and provided new images in **Supplementary Figure 6** and **7**. We also performed SOX9 staining in lung type II (AT2) cells and Pdgfra positive mesenchymal cells. As shown in **Supplementary Figure 7**, SOX9 was colocalized in lung AT2 cells, but not in Pdgfra positive cells. Sox9 is a stem cell associated transcription factor which plays multiple roles in the lung epithelium during branching morphogenesis. Neonatal hyperoxia significantly reduced the numbers of Sox9⁺ AT2 cells (**Supplementary Figure 7**). This suggests that neonatal hyperoxia dampens the ability of Sox9 to promote AT2 cell proliferation, leading to alveolar simplification.

As shown in immunofluorescent staining of an AT1 cell biomarker Hopx with lamin b1 or p21 (**Supplementary Figure 5**), there were no senescence in AT1 cells in the lung of mice exposed to hyperoxia as neonates. We administered supernatants from cultured senescent AT2 cells into neonatal mice at pnd4, and found this caused alveolar and vascular simplification at pnd14 (**Figure 6**). Thus, senescent AT2 cells contribute to hyperoxia-induced lung injury via secretory SASP factors.

As shown in **Figure 5e-h** and **Supplementary Figure 7**, neonatal hyperoxia caused senescence in Pdgfra mesenchymal cells, and reduced number of Pdgfra⁺ cells, in the lung at pnd7.

Point 5. *Can the authors comment on the mechanisms by which programmed senescence promotes normal alveologenesi s?*

Response: As shown in **Supplementary Figure 9g**, IL-10, TNF α , MMP-8, and PAI-1 protein levels were reduced, whereas levels of TNFSF12 and MMP-12 proteins were increased in the lung during postnatal lung development. This suggests that the changes of these SASP factors from programmed senescent cells may promote normal alveologenesi s. For instance, TNFSF12 (Tweak)

is a mitogenic and migration factor for endothelial cells and promotes angiogenesis (PMID: 12615668). Increased extracellular matrix degradation by MMP-12 may further create an environment which allows endothelial cell migration and proliferation induced by TNFSF1 for vascularization during lung development. We have discussed this in the revised manuscript.

Point 6: Figure 1C-the quality of the images is distinct; O2 exposed image has considerable background.

Response: As suggested, we have replaced Figure 1C in the revised manuscript.

Point 7: Figure 1E, F – The history, diagnosis, gestational and postnatal ages of the patients included in the analysis should be provided. Supplementary data showing histology of these lung sections are needed to understand regional pathology.

Response: As suggested, we have provided history, diagnosis, gestational and postnatal ages of the patients as well as showing histology of these lung sections in the revised manuscript (**Supplementary Figure 1** and **Supplementary Table 1**).

Point 8: Suggest moving Figure 2 to the Supplement or combining panels rather than multiple separate figures.

Response: As suggested, we have moved **Figure 2** into the **Supplementary Figure 2** in the revised manuscript.

Point 9: Figure 4F – The analysis, algorithms used, data quality, numbers of cells, and statistical analysis are needed to support the figure. Raw data should be made available, or the accession number should be provided if published. If data are from previously published and analyzed, this might be stated and referenced in the legend to clarify its origin. The specific cell types contributing to SASP (4F) should be indicated and statistically validated.

Response: As suggested, we have provided information of algorithms used, data quality, numbers of cells, and statistical analysis in the revised manuscript. We have deposited the scRNA-seq raw data in the Gene Expression Omnibus database under primary accession code GSE207866. In Figure 3g of the revised manuscript, some of data are derived from our recent report (PMID: 34417156). We have stated this in the figure legend. The specific cell types contributing

to SASP have been indicated and statistically validated in the revised manuscript (**Supplementary Figure 4a** and **Supplementary Data 1-8**).

Point 10: Figure 5 – Are AT1 cells also senescent? Which fibroblastic cell types?

Response: Although a few of senescent AT1 cells were identified through the scRNA-seq analysis (**Supplementary Figure 4a**), there were no senescent AT1 cells identified by immunofluorescence (**Supplementary Figure 5a-b**). We also performed the staining of Pdgfra with lamin b1 or p21, and found senescence in Pdgfra positive cells in the lung of mice exposed to hyperoxia at pnd7 (**Figure 5e-h**). This suggests that hyperoxia may cause senescence in Pdgfra positive secondary crest myofibroblasts in the lung. We have included this in the manuscript.

Figure 11: Figure 6 – SOX9 staining is diffuse. It is usually nuclear; there is high background. What is the cell type of the SOX9+ cells?

Response: As suggested, we have reperfomed SOX9 staining and provided new images in **Supplementary Figure 6**. We also performed SOX9 staining in lung AT2 cells and Pdgfra positive mesenchymal cells. As shown in **Supplementary Figure 7**, SOX9 signal was colocalized in lung AT2 cells, but not in Pdgfra positive cells. Neonatal hyperoxia significantly reduced the numbers of Sox9⁺ AT2 cells (**Supplementary Figure 7**). This suggests that neonatal hyperoxia dampens the ability of Sox9 to promote AT2 cell proliferation, leading to alveolar simplification.

Reviewer 2's comments:

Point 1: *Results. Fig.1F. Were the investigators not able to detect SA- β -gal in the human premature infants?*

Response: The SA- β -gal assay requires fresh but not formalin-fixed, paraffin-embedded samples. Currently, we do not have OCT embedded frozen lung blocks for SA- β -gal assay in human premature infants.

Point 2: *Results. Fig. 1H. Based on Edu staining only, the authors give a strong statement that lung senescence is reduced in the saccular stage. To prove this both Ki67 and EdU should be done because Ki67 detects all stages of cell division (except G0) while Edu labels only during the "S" phase.*

Response: As suggested, we have provided both EdU and Ki67 staining (**Figure 1g-h**) in the revised manuscript.

Point 3: *Results. Fig. 2. The authors have shown there is no sex differences in senescence, but only at 1 time point (pnd7). Other time points should have been shown for consistency and confirmation. This is important for translational significance in this experimental BPD mouse model, as it is well known that there is a preponderance of male infants in the development and severity of human BPD (for e.g. see PMIDs: 31727986; 24421662).*

Response: We are aware and have discussed that there is a preponderance of male infants in patients who develop BPD and a correlation of male sex with severity of human BPD (e.g. see PMIDs: 31727986; 24421662). As suggested, we have compared sex differences in lung senescence after hyperoxic exposure at pnd3, pnd7, pnd10 and pnd60 (**Supplementary Figure 2**). We have included these data in the revised manuscript.

Point 4: *Results. Fig. 3F. Interestingly, other authors have shown that p53 protein is increased in neonatal mouse lungs after hyperoxia on pnd3. See Fig. 6 in PMID: 21270677. How do the investigators explain their result of no difference in p53 protein results? Furthermore, the investigators show phospho-p53 at S18 in the mouse lung is increased and claim this suggests cellular senescence is increased. However, phosphorylation of p53 is well described to be*

increased in apoptosis in neonatal lungs exposed to hyperoxia (for e.g. see PMID: 11435201). Thus, the authors show an association, not causation, as regards senescence.

Response: In the referenced paper (PMID: 21270677), p53 was increased in the lung of FVB/n mice exposed to hyperoxia (80% O₂), whereas p53 protein levels were not changed in the lung of C57BL/6J mice exposed to hyperoxia (>95% O₂) in our study. This suggests that hyperoxia-induced change in p53 protein expression may be dependent on strains and oxygen concentrations. Indeed, FVB/n mice are more susceptible to developing neonatal hyperoxia-induced septal thickness and alveolar simplification compared to the C57BL/6J strain (PMID: 30848059).

In a review article (PMID: 11435201), the authors suggest that phosphorylation of p53 is increased in apoptosis in neonatal lungs exposed to hyperoxia. To answer the questions as to whether phosphorylation of p53 modulates hyperoxia-induced apoptosis in the lung, we performed dual immunofluorescence to co-localize phosphorylated p53 (ser 18) with cleaved caspase-3 or annexin V in the lung of mice exposed to hyperoxia at pnd7. As shown in **Supplementary Figure 3e-h**, we did not see colocalization of phosphorylated p53 (ser 18) with cleaved caspase-3 or annexin V. This suggests that phosphorylated p53 at ser 18 may not directly cause apoptosis in the lung of neonatal mice exposed to hyperoxia. Thus, we did not pursue further studies on the causal relationship between phosphorylated p53 (ser 18) and neonatal hyperoxia-induced apoptosis.

We agree with the comment that there is an association, not causation, of phosphorylated p53 (ser 18) with neonatal hyperoxia-induced senescence. A study using transgenic mice with a mutated S18 at p53 (Trp53^{tm2Xu}, Cat#006980) (PMID:16757976) would answer the question on the role of phosphorylated p53 (ser 18) in regulating neonatal hyperoxia-induced senescence. However, this strain of mice is cryopreserved in the Jackson Laboratory, which will take approximate 4 months for recovery. Hence, we are unable to provide specific data on a causal relationship between phosphorylated p53 (ser 18) and neonatal hyperoxia-induced senescence due to the limited time window for revising this manuscript. Nevertheless, our recent report showed that p53 deletion inhibits hyperoxia-induced senescence in cultured lung epithelial cells (PMID: 34042288). This suggests that p53 might contribute to neonatal hyperoxia-induced senescence in the lung. We have discussed this in the revised manuscript.

Point 5: *Results. Fig. 4D. 8-oxo-DG has been shown to be increased in hyperoxia-exposed*

neonatal rats in association with apoptosis (see PMID: 32933455).

Response: 8-oxo-DG is one of the most widely studied oxidized metabolites and is considered as a biomarker for oxidative damage of DNA. We agree with the comment that 8-oxo-DG has been shown to be increased in hyperoxia-exposed neonatal rats in association with apoptosis (see PMID: 32933455). The levels of 8-oxo-DG were also increased in senescent cells (PMID: 7753808, 32047109). Thus, as a biomarker for oxidative damage of DNA, 8-oxo-DG could participate in neonatal hyperoxia-induced apoptosis and senescence. We have discussed this in the revised manuscript.

Point 6: Results. Fig. 4F. Senescence leading to inflammation and the presence of inflammatory markers have not been confirmed by protein data.

Response: As suggested, we have provided protein levels of inflammatory markers using a Luminex assay (**Figure 3f**) in the revised manuscript.

Point 7: The selective p21 inhibitor (UC2288) is also known to attenuate apoptosis, and an “alternative method by which p21 might protect against hyperoxia is through its ability to block apoptosis” (PMID: 16723699). How do the authors reconcile the blocking of the apoptosis pathway (versus senescence) in terms of improving the lung morphometry?

Response: In this paper using adult p21 deficient mice and WT littermates (PMID: 16723699), p21 protects against hyperoxia by maintaining expression of anti-apoptotic Bcl-X_L protein. In the present study, we used a selective p21 inhibitor (UC2288) and found that UC2288 inhibited hyperoxia-induced senescence and alveolar simplification in neonatal mice. Thus, UC2288 treatment might inhibit neonatal hyperoxia-induced alveolar simplification by decreasing both apoptosis and senescence. In addition, senescent cells are resistant to apoptosis. The effects of UC2288 on apoptosis and senescence may be cell specific. We have discussed this in the revised manuscript.

Point 8. As also mentioned by the authors (ref.# 45), p21 null mutant neonatal mice have been described in a 2004 report to have worse lung morphometry upon hyperoxia-exposure.

Response: We are aware that six weeks old p21 null mutant neonatal mice (129/Sv × C57BL/6 background) have worse lung morphometry after a 4-day neonatal hyperoxic exposure compared

to wild-type littermates (PMID: 14607813). In the present study, we evaluated these mutant mice in normoxic environments from pnd0 and found that neonatal p21 knockout mice with a C57BL/6 background have disrupted lung architecture from birth. This may be the cause of the susceptibility to hyperoxia-induced lung injury described previously (PMID: 14607813). We have discussed this in the revised manuscript.

***Point 9.** Incidentally, quercetin alone has been used in neonatal mice exposed to hyperoxia and led to attenuation of hyperoxia-induced lung injury. Please see PMID: 29432836.*

Response: We thank the reviewer for bringing this to our attention. In this paper (PMID: 29432836), treatment with quercetin (20 mg/kg, i.p.) attenuated hyperoxia-mediated alveolar simplification in mice. This further confirms our findings that low doses of quercetin (2.5 and 5 mg/kg, i.p.) in combination with dasatinib inhibited neonatal hyperoxia-induced alveolar and vascular simplification in mice. This combination of the two agents may also reduce possible side effects of quercetin administered at a high dose of 20 mg/kg. In addition, quercetin (20 mg/kg, i.p.) reduced hyperoxia-induced inflammatory responses (PMID: 29432836), while a combination of low doses of quercetin with dasatinib inhibited neonatal hyperoxia-induced lung senescence. As shown in **Figure 3f**, senescent cells are able to produce SASP factors including inflammatory mediators. Thus, quercetin alone may attenuate hyperoxia-induced lung injury by decreasing senescence and SASP factors. We have discussed this in the revised manuscript.

***Point 10.** Surprisingly, there is no description of the impact of the above-mentioned manipulations on the vascular phenotype in the experimental BPD mouse model.*

Response: We thank the reviewer for bringing this to our attention. In the revised manuscript, we have provided data on the vascular phenotype by measuring vWF positive vessel numbers in our experimental BPD mouse model.

Reviewer 3's comments:

***Point 1:** The manuscript by Yao et al (NCOMMS-21-42376) describes the identification of developmentally-regulated senescent cells in the mouse lung at birth that decrease throughout the saccular stage. In addition, exposure of newborn mice to hyperoxia at pnd3 led to increased senescence, mediated apparently by p53/p21 in mesenchymal and type II cells. Eliminating these O₂-induced senescent cells with the senolytic cocktail of dasatinib + quercetin (D+Q) during the alveolar stage inhibited hyperoxia-induced alveolar simplification. The authors conclude programmed senescence orchestrates postnatal lung development whereas later hyperoxia-induced senescence causes alveolar simplification. These results are of potential interest since they suggest that senolytic treatment during the alveolar stage could be used therapeutically to prevent hyperoxia-induced alveolar simplification. However, much of the analysis is based on immunostaining, supported by scRNA seq in only one type of experiment. The authors need to provide further evidence that at least two markers of senescence (e.g., p21+, Lamin B-, HBGB1-, p53BP1 foci, TAFs, etc) co-localized to the same cell. Also, the authors need to characterize the developmentally regulated senescence at pnd0 in further detail in regard to their senescent signature compared to hyperoxia-induced senescence.*

Response: We thank the reviewer for favorable comment that these results are of potential interest since senolytic treatment during the alveolar stage could be used therapeutically to prevent hyperoxia-induced alveolar simplification. As suggested, we have performed co-localization of p21 or phospho-p53 with lamin b1, and found that p21 or phospho-p53 positive cells lacked lamin b1 in the lung of mice exposed to hyperoxia as neonates at pnd7 (**Supplementary Figure 3a-d**). In addition, we showed that neonatal hyperoxia increased p21 signal and loss of nuclear lamin b1 in pro-SPC⁺ and mesenchymal (Pdgfra⁺) cells (**Figure 4, 5**). The scRNA-seq data also demonstrated senescence in AT2 and mesenchymal cells in mice exposed to hyperoxia as neonates (**Supplementary Figure 4a**). Altogether, neonatal hyperoxia causes senescence in AT2 and Pdgfra⁺ mesenchymal cells.

As suggested, we have characterized the developmentally regulated senescence in further detail in regard to their senescent signature compared to hyperoxia-induced senescence in terms of SASP factors, DNA damage and cell types. Generally, SASP factors are different between the developmental and hyperoxia-induced senescence (**Supplementary Figure 9g**). In addition, and

in contrast with hyperoxia induced senescence, there was no change in DNA damage during postnatal lung development (**Supplementary Figure 9a-f**). In early postnatal development, we detected senescence in vimentin⁺ mesenchymal cells, whereas there was no senescence in AT2 or Pdgfra⁺ cells during postnatal lung development, in contrast to hyperoxia induced senescence (**Supplementary Figure 10**). These findings suggest different mechanisms responsible for developmentally regulated and hyperoxia-induced senescence in the lung with the developmental senescence leading to the thinning of the mesenchyme whereas hyperoxia-induced senescence resulting in alveolar septal blunting and loss of type II cells.

***Point 2:** The authors need to co-localize their senescent cell markers to the same cell. For example, are p21⁺ cells always Lamin B or HMGB1 low and p53BP1 positive?*

Response: As suggested, we have performed co-localization of p21 and lamin b1, and found that p21 positive cells lacked lamin b1 in the lung of mice exposed to hyperoxia as neonates at pnd7 (**Supplementary Figure 3a**), identifying these cells as senescent.

***Point 3:** It is somewhat surprising that qPCR and bulk RNAseq analysis did not reveal an increase in SASP factors. Although this could be due to low number of senescent cells, p21 is highly upregulated at the expression level. Does this mean the senescent cells in the lung only express a low level of SASP?*

Response: In the revised manuscript, we have provided the SASP factor protein levels in the lung of mice exposed to hyperoxia as neonates (**Figure 3f**). Although the expression of these genes in whole lung homogenates was not significantly increased by neonatal hyperoxia, the protein levels of SASP factors were increased in the lung of mice exposed to hyperoxia as neonates. These data suggest that senescent cells in the lung can secrete SASP factors.

***Point 4:** It is hard to visualize in the images provided what the authors are calling positive or negative for immunofluorescence staining. Larger images should be included as supplemental figures.*

Response: As suggested, we have provided larger images used in **Figure 1-5** as supplementary figures in the revised manuscript.

Point 5: *The analysis of p53 phosphorylation by Elisa, especially with the small increases observed, isn't convincing. Can Western analysis of p53 phosphorylation in the C12FDG positive cells be performed? Alternatively, p21 and phospho-p53 can be co-localized by CyTOF.*

Response: As suggested, we have provided the data on p53 phosphorylation in the C12FDG positive cells of mice exposed to hyperoxia in the revised manuscript (**Figure 2g**).

Point 6: *The p21 inhibitor UC2288 supposedly decreases p21 mRNA expression independently of p53. Does this mean that the upregulation of p21 in senescent cells is independent of p53?*

Response: We used the p21 inhibitor UC2288 to block p21-mediated senescence and determine its effect on neonatal hyperoxia-induced alveolar simplification. Since the total abundance of p53 was unchanged while p21 was increased in the lung of hyperoxia-exposed mice (**Figure 2**), we did not evaluate the total levels of p53 after UC2288 treatment. It has been shown that phosphorylation event of p53 at S18 can increase its recruitment on the p21 promoter, thereby augmenting p21 gene expression (PMID: 24928858). The phosphorylation of p53 at S18 was increased in mice exposed to hyperoxia (**Figure 2**). Therefore, our results suggest that increased phosphorylation of p53 at S18 may lead to hyperoxia-induced p21 gene expression in the lung.

Point 7: *Is there disrupted lung development in p53 KO mice?*

Response: We are not sure whether lung development is disrupted in p53 KO mice. However, a previous report showed that p53 KO mice are viable and appear completely normal but are susceptible to the spontaneous development of different types of tumors (PMID: 1552940). Since no changes of total p53 but rather increased phosphorylated p53 (ser15) were observed in the lung of mice exposed to hyperoxia, the study using transgenic mice with a mutated S18 at p53 (Trp53^{tm2Xu}) (PMID:16757976) would answer the question on the role of phosphorylated p53 (ser 18) in modulating lung development. However, this strain of mice is cryopreserved in the Jackson Laboratory (Cat#006980), which will take more than 4 months for recovery of the strain. Hence, we are unable to provide the data on causal relationship of phosphorylated p53 (ser 18) with lung development due to the limited time window for revising this manuscript.

Point 8: *The authors attempt to characterize the hyperoxia induced senescent cells, but not the ones elevated at pnd0. Do these developmentally regulated senescent cells express a similar SASP*

as the hyperoxia induced senescent cells? Do they express markers of DNA damage? What type of cells are senescent? Do they also express Sox9?

Response: As shown in **Supplementary Figure 9g**, the profile of lung SASP factors is different between developmentally regulated and hyperoxia-induced senescence (at pnd7). There were no changes in 53BP1 or γ H2AX expression during postnatal lung development (**Supplementary Figure 9a-f**), suggesting no DNA damage involved in developmental senescence. As shown in **Supplementary Figure 10**, during postnatal lung development, we observed senescence in vimentin positive cells but not in AT2 or Pdgfra positive cells, as seen in hyperoxic senescence. Further study using scRNA-seq is warranted to identify which type of vimentin positive cells are senescent during lung development. In the mouse lung, Sox9 protein levels disappear in the first day after birth (PMID: 24274029). Thus, we do not think that developmentally programmed senescent cells express Sox9 since these senescent cells persist until day 3.

***Point 9:** There are preliminary reports of the senolytic combination of D+Q not killing certain “beneficial” senescent cells so the fact that these cells appear to be cleared by D+Q is of interest. It is also of interest since clinical trials to treat eclampsia are being planned. Thus, these developmentally-regulated senescent cells need to be characterized further.*

Response: As suggested, we have characterized developmentally regulated senescent cells as shown in responses to the point 8 raised by this reviewer. In the present study, the senolytic combination dasatinib and quercetin when administered early (pnd1-3) eliminated the beneficial/developmentally regulated senescent cells and deleteriously altered lung morphology. In contrast, removing the hyperoxia-induced senescent cells by administration of dasatinib/quercetin at pnd4-6, once injury had occurred, improved lung morphology. This demonstrates that **timing** of administration to minimize elimination of beneficial (developmental) senescent cells is extremely important in dictating a beneficial vs detrimental effect. It is not clear how this would be extrapolatable to placental senescence but this would be intriguing to explore. We have discussed this in the revised manuscript.

In summary, we are very grateful to the reviewers for their insightful comments. The manuscript is now much clearer and more mechanistic. We look forward to a favorable response as to its acceptance.

Reviewer comments, second round

Reviewer #1 (Remarks to the Author):

The authors have carefully considered comments from all three reviewers and have added the requested experimental data and clarifications. Likewise, concerns regarding the need for better colocalization of markers and poor quality of immunofluorescence and light microscopy were addressed. The work demonstrates distinct developmental roles for senescence, supporting its beneficial role in the normal perinatal lung process and its potential damaging role during hyperoxia-induced alveolar simplification. The remaining minor concerns that the author might consider:

1. The abstract and results state that “eliminating senescence cells” yet in the results, e.g., on line 258 the authors suggested senolytics or p21 inhibitors “clear cells”. Since “clearance” or “removal of senescence cells” is inferred in all of these experiments and not formally shown, perhaps the authors precisely describe their findings, e.g. treatment with senolytics, decreased expression of senescence markers, and restored alveolarization, etc.

Monir concerns:

1. “Clearance” removal is used without definitive data that the cells are removed, only that these Q/D and p21 inhibitors inhibited senescence markers. The authors strongly state that p16 is not involved in alveolarization or repair which is a rather strong statement based on the lack of changes in p16 RNA.
2. Line 198, Regarding co-expression of macrophage in AT2 cell markers, were the sc-RNA data corrected for ambient RNA? The authors have not shown that macrophages engulf senescence type II cells, although they may also take up apoptotic or necrotic type II cells or their products.

Reviewer #2 (Remarks to the Author):

General

This manuscript describes the detection of senescence during lung development and in response to hyperoxia-induced injury. Lung senescence is increased at birth and decreases subsequently. The investigators show removing senescent cells during the sacular stage disrupted lung development. During hyperoxia-exposed lung injury, lung senescence is increased in Type II cells and secondary crest myofibroblasts, which peaked during the alveolar stage. Hyperoxia activates the p53/p21 (but not the p16) pathway, leading to cellular senescence. Senescent cells contribute to hyperoxia-induced alveolar simplification and vascular dysregulation through secretory SASP factors. Removing senescent cells at this stage in this lung-injury model was protective. Data on using the senolytic cocktail (quercetin and dasatinib) during the sacular and alveolar stage of lung development is provided. In addition, the investigators have used the p21 null mutant neonatal mouse and a selective p21 inhibitor to study the role of senescence in postnatal lung development and after hyperoxia exposure. The importance of the lung developmental stage and timing of senescence in normal and abnormal (hyperoxia-induced) scenarios in neonatal mice have been elegantly highlighted in this manuscript. This has potential translational significance for human premature neonates developing bronchopulmonary dysplasia.

Specific

The authors have appropriately responded to all my specific queries in the revised manuscript.

Reviewer #4 (Remarks to the Author):

Authors investigated to find an optimal therapeutic window for mitigating neonatal hyperoxia lung injury by clearing senescence. They tried to verify two phenomena, one is development senescence in neonatal lung, another is hyperoxia cell death in sacular stage. Authors we showed that early programmed senescence orchestrates postnatal lung development, while neonatal hyperoxia-induced senescence causes lung alveolar and vascular simplification. This manuscript is timely and interesting, however, there are many concerns in methods and interpretation of data.

Point1. In Figure 1, IHC in Figure 1a showed almost all cells are not stained for Laminb1, which is not compatible to Fig1b.

Point 2. In supplementary Figure 4a, the scRNA-seq data in C12FDG sorted cells at pnd7 showed most of cells are macrophages, and type II cells are few. These results do not seem to be reliable. Authors should address the role of these senescent macrophages, although they suggest that certain macrophages engulf and ingest senescent type II cells in the lung of mice exposed to hyperoxia as neonates.

Point 3. In Figure 4, LaminB1 stain looks non-specific background.

Point 4. In Figure 5, staining for lamin B1 and Vimentin are not clear and non-specific. Staining for vimentin was variable. Cells positive for PDGFra do not look like myofibroblast. It is questionable to quantitate lamin B1 loss in this IHC staining.

Point 5. In supplementary Figure 6, it should be addressed to examine the percentage of SOX9 positive type II cells in senescent type II cells in order to understand the significance of increased SOX9 expression.

Point 6. In Figure 6, it is interesting that culture supernatant of senescent type II cells induced remodeling of instilled lung. However, authors should address what components of supernatant could induce lung remodeling, and also address how much percentage of lung cells are senescent.

Manuscript #: NCOMMS-21-42376A

Timing and cell specificity of senescence drives postnatal lung development and injury

Yao et al.

[redacted]

We are confident that the manuscript is greatly improved and look forward to a favorable response.

Reviewer #1 Comments:

Point 1: The authors have carefully considered comments from all three reviewers and have added the requested experimental data and clarifications. Likewise, concerns regarding the need for better colocalization of markers and poor quality of immunofluorescence and light microscopy were addressed. The work demonstrates distinct developmental roles for senescence, supporting its beneficial role in the normal perinatal lung process and its potential damaging role during hyperoxia-induced alveolar simplification. The remaining minor concerns that the author might consider.

Response: We thank the reviewer for the favorable comments that we have carefully considered comments from all three reviewers, added the requested experimental data and clarifications, and addressed the need for better colocalization of markers and poor quality of immunofluorescence and light microscopy in the first version. We are also pleased to hear that our work demonstrates distinct developmental roles for senescence, supporting its beneficial role in the normal perinatal lung process and its potential damaging role during hyperoxia-induced alveolar simplification.

Point 2: The abstract and results state that “eliminating senescence cells” yet in the results, e.g., on line 258 the authors suggested senolytics or p21 inhibitors “clear cells”. Since “clearance” or “removal of senescence cells” is inferred in all of these experiments and not formally shown, perhaps the authors precisely describe their findings, e.g. treatment with senolytics, decreased expression of senescence markers, and restored alveolarization, etc.

Response: As suggested, we have revised manuscript by providing a more precise description of what the experiments show in the revised manuscript.

Point 3: *“Clearance” removal is used without definitive data that the cells are removed, only that these Q/D and p21 inhibitors inhibited senescence markers. The authors strongly state that p16 is not involved in alveolarization or repair which is a rather strong statement based on the lack of changes in p16 RNA.*

Response: As requested, we have more precisely described the findings of our experiments by stating that “senescence markers are inhibited” rather than “senescent cells are cleared” in the revised manuscript. We have also softened the statement regarding conclusions from our p16 data in the revised manuscript in view of the reviewer’s comments.

Point 4: *Line 198, Regarding co-expression of macrophage in AT2 cell markers, were the sc-RNA data corrected for ambient RNA? The authors have not shown that macrophages engulf senescence type II cells, although they may also take up apoptotic or necrotic type II cells or their products.*

Response: We thank the reviewer for this important comment that macrophage may also take up apoptotic or necrotic type II cells or their products rather than strictly engulf senescent Type II cells. We have addressed this in the revised manuscript.

Reviewer 2's Comments:

Point 1: The importance of the lung developmental stage and timing of senescence in normal and abnormal (hyperoxia-induced) scenarios in neonatal mice have been elegantly highlighted in this manuscript. This has potential translational significance for human premature neonates developing bronchopulmonary dysplasia. The authors have appropriately responded to all my specific queries in the revised manuscript.

Response: We thank this reviewer for the favorable comments and the lack of further critiques. We are thankful that we satisfied the concerns raised previously.

Reviewer 4's comments:

***Point 1:** Authors investigated to find an optimal therapeutic window for mitigating neonatal hyperoxia lung injury by clearing senescence. They tried to verify two phenomena, one is development senescence in neonatal lung, another is hyperoxia cell death in saccular stage. Authors we showed that early programmed senescence orchestrates postnatal lung development, while neonatal hyperoxia-induced senescence causes lung alveolar and vascular simplification. This manuscript is timely and interesting, however, there are many concerns in methods and interpretation of data.*

Response: We thank the reviewer for the favorable comment that our study is timely and interesting. We have addressed the concerns as shown below.

***Point 2:** In Figure 1, IHC in Figure 1a showed almost all cells are not stained for Laminb1, which is not compatible to Fig1b.*

Response: The Figure 1a and 1b show SA- β -gal staining, whereas lamin b1 staining is shown in Figure 1c and 1d. I believe the reviewer is referring to SA- β -gal staining. In Figure 1b we counted SA- β -gal positive cells with weak, moderate and strong signals, whereas the arrows show the strong SA- β -gal signal in Figure 1a. To avoid the confusion, we have pointed out all positive cells stained with SA- β -gal in larger images in the Supplemental Figures.

***Point 3.** In supplementary Figure 4a, the scRNA-seq data of C12FDG sorted cells at pnd7 showed that most of cells are macrophages, and type II cells are few. These results do not seem to be reliable. Authors should address the role of these senescent macrophages, although they suggest that certain macrophages engulf and ingest senescent type II cells in the lung of mice exposed to hyperoxia as neonates.*

Response: We thank the reviewer for the comments on the role of senescent macrophages. As suggested, we further analyzed the scRNA-seq data on C12FDG sorted macrophages. We identified six Seurat clusters, including clusters 13, 15, 16, 17, 21, and 23. Macrophages may change their polarization status once they acquire a senescence-like phenotype or under exposure of the SASP factors (PMID: 33355620). Thus, we wanted to evaluate the polarization phenotype of these clusters. As shown in Supplementary Data 4, clusters 13 and 15 highly expressed M2

markers, while cluster 23 exhibited a mixed M1/M2 phenotype. M2 macrophages usually have high phagocytosis capacity. Indeed, clusters 13 and 15 of macrophages expressed both M2 markers and phagocytosis regulatory genes (Supplementary Data 3, 5, 6), suggesting these subpopulations are polarized into M2 status for phagocytosis. Interestingly, senescent macrophages can also be polarized into M2 phenotype, and activate a pro-fibrotic transcriptional program (PMID: 28768895 and 35181634). Further study is warranted to determine whether these subpopulations also play important roles in mediating lung fibrogenesis in hyperoxia-exposed mice. We have discussed it in the revised manuscript.

Point 4. In Figure 4, Lamin B1 stain looks non-specific background.

Response: We have selected better images of lamin b1 and SPC for Figure 4. We apologize for the lack of clarity in the previous version.

Point 5. In Figure 5, staining for lamin B1 and Vimentin are not clear and non-specific. Staining for vimentin was variable. Cells positive for PDGFRA do not look like myofibroblast. It is questionable to quantitate lamin B1 loss in this IHC staining.

Response: As suggested, we have replaced the images in Figure 5 of the revised manuscript with clearer ones.

Point 6. In supplementary Figure 6, it should be addressed to examine the percentage of SOX9 positive type II cells in senescent type II cells in order to understand the significance of increased SOX9 expression.

Response: We thank this reviewer for this thoughtful comment. Due to the technical difficulties of triple staining of SOX9, SPC, and lamin b1 (or p21), we are unable to calculate the percentage of SOX9 positive senescent type II cells in the lung with immunohistochemistry. Alternatively, we compared SOX9 gene expression between senescent type II cells isolated from hyperoxia-exposed mice and non-senescent type II cells isolated from air control mice at pnd7. We found that SOX9 gene expression exhibited a 58% reduction in senescent type II cells compared to those in non-senescent type II cells (Supplementary Figure 6g). We have incorporated it in the revised manuscript.

Point 7. In Figure 6, it is interesting that culture supernatant of senescent type II cells induced remodeling of instilled lung. However, authors should address what components of supernatant could induce lung remodeling, and also address how much percentage of lung cells are senescent.

Response: As shown in Figure 6b, the levels of IL-1 α , IL-1 β , Cxcl2, Cxcl12, and Tnfrsf1b were significantly increased in the supernatants of senescent type II cells compared to those in non-senescent type II cells from air control. These SASP factors could be important components to induce lung remodeling and injury. Indeed, blocking IL-1 or Cxcl2 receptors prevents hyperoxic lung injury (PMID: 15004193, 23946428).

SASP factors may induce various cellular processes, including inflammatory responses, apoptosis or cell death, leading to lung remodeling and injury. Thus, we did not calculate the percentage of lung cells that are senescent after instillation of culture supernatant from senescent type II cells. Nevertheless, we have discussed it in the revised manuscript.

Reviewer comments, third round

Reviewer #1 (Remarks to the Author):

This is a revised manuscript providing new insights into the role of senescence in normal lung morphogenesis and its role in mediating alveolar simplification during hyperoxia exposure in mice. Data are complete and support their conclusions that cell type-specific senescence is a critical factor in normal lung maturation and alveolar loss during hyper-exposure. Decreasing senescence cells improved lung structure during toxicity caused by hyperoxia.

Reviewer #4 (Remarks to the Author):

This study is interesting, and authors responded to my comments sufficiently. Although there are several experiments which should be performed using single cell analysis, authors have done a lot of IHC and presented reliable results.

Manuscript #: NCOMMS-21-42376B

Timing and cell specificity of senescence drives postnatal lung development and injury

Yao et al.

[redacted]

We are confident that the manuscript is greatly improved and look forward to a favorable response.

Reviewer #1's comments:

Point 1: This is a revised manuscript providing new insights into the role of senescence in normal lung morphogenesis and its role in mediating alveolar simplification during hyperoxia exposure in mice. Data are complete and support their conclusions that cell type-specific senescence is a critical factor in normal lung maturation and alveolar loss during hyper-exposure. Decreasing senescence cells improved lung structure during toxicity caused by hyperoxia.

Response: We are glad to hear that data are complete and support the conclusion.

Reviewer #4's comments:

Point 1: This study is interesting, and authors responded to my comments sufficiently. Although there are several experiments which should be performed using single cell analysis, authors have done a lot of IHC and presented reliable results.

Response: We thank the reviewer for the favorable comment that the study is interesting and that we have presented reliable results and responded to the comments sufficiently. We will continue to explore the scRNA-seq and conduct new analyses in the future to continue to define how senescence manifests in the newborn lung.